# FloydNet: A Learning Paradigm for Global Relational Reasoning

## Abstract

Developing models capable of complex, multi-step reasoning is a central goal in artificial intelligence. While representing problems as graphs is a powerful approach, Graph Neural Networks (GNNs) are fundamentally constrained by their message-passing mechanism, which imposes a local bottleneck that limits global, holistic reasoning. We argue that dynamic programming (DP), which solves problems by iteratively refining a global state, offers a more powerful and suitable learning paradigm. We introduce FloydNet, a new architecture that embodies this principle. In contrast to local message passing, FloydNet maintains a global, all-pairs relationship tensor and learns a generalized DP operator to progressively refine it. This enables the model to develop a task-specific relational calculus, providing a principled framework for capturing long-range dependencies. Theoretically, we prove that FloydNet achieves 3-WL (2-FWL) expressive power, and its generalized form aligns with the k-FWL hierarchy. FloydNet demonstrates state-of-the-art performance across challenging domains: it achieves near-perfect scores (often >99%) on the CLRS-30 algorithmic benchmark, finds exact optimal solutions for the general Traveling Salesman Problem (TSP) at rates significantly exceeding strong heuristics, and empirically matches the 3-WL test on the BREC benchmark. Our results establish this learned, DP-style refinement as a powerful and practical alternative to message passing for high-level graph reasoning.

## 1 Introduction

The development of intelligent models capable of reasoning is a great challenge in the field. Modern approaches can be analyzed through the lens of dual process theory (Kahneman, 2011), which distinguishes between intuitive "System 1" thinking and deliberate, step-by-step "System 2" thinking. While Large Language Models (LLMs) (Google, 2025; OpenAI, 2025; Deepseek, 2025) exhibit impressive emergent reasoning akin to System 1 (Wang & Zhou, 2024), ensuring structured and verifiable logic, the hallmark of System 2, remains an open problem. A key step towards building such reasoners is learning to model the complex, global relationships between representations, not just their individual features.

Neural Algorithmic Reasoning (NAR) (Veličković & Blundell, 2021) provides a promising framework for this challenge, aiming to bridge the logic of classical algorithms with the power of deep learning. Graph Neural Networks (GNNs), particularly the Message Passing (MPNN) paradigm (Gilmer et al., 2017), have become a cornerstone of NAR. However, this architectural choice introduces a fundamental bottleneck: local message passing is inherently insufficient for problems demanding global, holistic reasoning. This locality manifests in two key limitations: **restricted information flow**, where capturing long-range dependencies requires deep GNNs that are prone to over-smoothing and over-squashing (Li et al., 2018; Topping et al., 2022), hindering effective scaling; and **limited expressive power**, with most MPNNs being bounded by the 1-WL test (Xu et al., 2019). Although graph transformers (GTs) (Dwivedi & Bresson, 2021; Ying et al., 2021) offer global receptive fields, they often rely on complex positional or structural encodings (Ying et al., 2021; Rampášek et al., 2022) to inject inductive bias into the graph, which can be difficult to design and may still underuse the higher-order relational structure of the graph.

In this work, we propose a new paradigm inspired by Dynamic Programming (DP). Instead of enriching node features locally, we learn a global refinement operator that acts directly on a representation of all-pairs relationships. The classic Floyd–Warshall (FW) algorithm exemplifies this principle: it

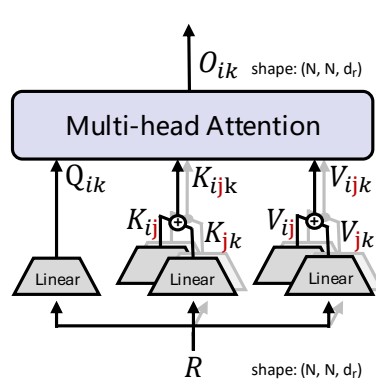

```python
def PivotAttnCore(q_ik, k_ij, k_jk, v_ij, v_jk):
    # All input tensors of shape [b,n,n,h,d_h]
    # come from different linear transform of R

    k_ijk = k_ij[:, :, :, None, :, :] +
            k_jk[:, None, :, :, :, :]
    v_ijk = v_ij[:, :, :, None, :, :] +
            v_jk[:, None, :, :, :, :]
    q_ixk = q_ik[:, :, None, :, :, :]

    a_ijk = torch.sum(q_ixk * k_ijk, dim=-1)
    d = q_ik.shape[-1] # per head embed dim
    a_ijk = a_ijk / (d ** 0.5) # [b,n,n,n,h]
    w_ijk = torch.softmax(a_ijk, dim=-3) # dim j

    o_ijk = v_ijk * w_ijk[:, :, :, :, :, None]
    o_ik = torch.sum(o_ijk, dim=-4) # dim j
    return o_ik # [b,n,n,h,d_h], same as input
```

Figure 1: **FloydBlock and Pivotal Attention Mechanism.** Each `FloydBlock` updates the representation for a given pair $(i, k)$ by using its state as a query to attend to all possible two-hop paths that pass through an intermediate "pivot" node $j$.

updates all-pairs shortest paths by systematically considering each node $j$ as a potential intermediate pivot in the path from $i$ to $k$. This iterative, all-pairs update allows information to propagate globally and interact in a structured way. Our key insight is to generalize this algorithmic structure. We replace the fixed update rule of FW (e.g., $D_{ik} \leftarrow \min_j(D_{ij} + D_{jk})$) with a learned, high-dimensional neural operator, creating an architecture that is algorithmically aligned with the global reasoning pattern of DP.

This principle is embodied in our new architecture, **FloydNet**, which maintains and refines a global pairwise relationship tensor $\boldsymbol{R} \in \mathbb{R}^{N \times N \times d_r}$. Its core component, the `FloydBlock`, acts as a *learned, generalized DP operator*. For each pair $(i, k)$, it updates its representation $\boldsymbol{R}_{ik}$ by aggregating information from all relational paths of the form $i \rightsquigarrow j \rightsquigarrow k$, mediated by every pivot node $j$. This update is not a fixed operation but a flexible transformation learned by our proposed **Pivotal Attention** mechanism. This data-driven approach allows FloydNet to learn a task-specific "relational calculus" and discover effective reasoning strategies beyond predefined heuristics.

Our primary contributions are:

1. We propose a new paradigm for graph reasoning that shifts from local message passing to a global, DP-inspired refinement of all-pairs relationships.

2. We introduce FloydNet, a novel architecture that realizes this paradigm through our proposed Pivotal Attention mechanism. We have also developed a highly optimized compute kernel of Pivotal Attention, which is efficient in processing large-scale graphs.

3. We provide extensive empirical validation, showing that FloydNet establishes new state-of-the-art results across a diverse range of challenging benchmarks, from algorithmic reasoning to combinatorial optimization.

## 2 THE FLOYDNET ARCHITECTURE

At its core, FloydNet is designed to implement the principle of learning a generalized Dynamic Programming operator. Instead of operating on node features, it maintains and iteratively refines a global pairwise relationship representation, $\boldsymbol{R} \in \mathbb{R}^{N \times N \times d_r}$. As illustrated in Figure 2, the architecture consists of three main stages: initialization of the relationship tensor, iterative refinement through a stack of `FloydBlock` layers, and a final decoder for task-specific representation learning.

## 2.1 Initialization of the Pairwise Relationship Representation

The initial all-pairs relationship tensor, $\boldsymbol{R}^{(0)}$, is constructed by integrating node, edge, and graph-level features into a unified relational representation. Given node features $\mathbf{X} \in \mathbb{R}^{N \times d_n}$, optional edge features $\mathbf{E} \in \mathbb{R}^{N \times N \times d_e}$, and optional graph features $\mathbf{G} \in \mathbb{R}^{d_g}$, we initialize the representation for each pair $(i, j)$ as:

$$\mathbf{R}_{ij}^{(0)} = \text{MLP}_{\text{init}}\Big( \big[ \mathbf{G}, \mathbf{X}_i, \mathbf{X}_j, \mathbf{E}_{ij} \big] \Big) \tag{1}$$

where $[\cdot, \cdot]$ denotes concatenation.

## 2.2 Adapting for Node and Graph-Level Tasks via a SuperNode

To handle node and graph-level tasks, we introduce a *learnable* **SuperNode (SN)**, a virtual node that acts as a global information aggregator.

- For Node-Level Tasks, the feature for node $i$ is its learned relationship with the SuperNode: $\mathbf{h}_i = \boldsymbol{R}_{i,\text{SN}}^{(L)}$.
- For Graph-Level Tasks, the graph representation is the SuperNode's self-relationship: $\mathbf{h}_{\text{graph}} = \boldsymbol{R}_{\text{SN,SN}}^{(L)}$.

As illustrated in Figure 2B, this provides a unified mechanism, allowing the model to learn optimal, task-specific aggregation functions through the same Pivotal Attention mechanism. This is achieved with negligible additional cost, increasing the graph size $N$ by only 1.

## 2.3 The FloydBlock: A Learned DP Operator

The central component of FloydNet is the `FloydBlock`, as shown in Figure 2. It takes a relationship tensor $\boldsymbol{R}^{(l-1)}$ and produces a refined tensor $\boldsymbol{R}^{(l)}$. A stack of $L$ such blocks is used to learn relational representation progressively. Each block follows a standard pre-LN Transformer structure, consisting of our Pivotal Attention module and a Feed-Forward Network (FFN):

$$\boldsymbol{R}' = \boldsymbol{R}^{(l-1)} + \text{PivotalAttention}(\text{Norm}(\boldsymbol{R}^{(l-1)})) \tag{2}$$

$$\boldsymbol{R}^{(l)} = \boldsymbol{R}' + \text{FFN}(\text{Norm}(\boldsymbol{R}')) \tag{3}$$

where Norm can be LayerNorm, RMSNorm (Zhang & Sennrich, 2019), or QK-Norm (Dehghani et al., 2023).

### 2.3.1 Pivotal Attention: The Core Mechanism

The Pivotal Attention mechanism, detailed in Figure 1, is the core of the `FloydBlock`. It updates the relationship representation for a target pair $(i, k)$ by aggregating information from all possible two-hop relational paths of the form $i \rightsquigarrow j \rightsquigarrow k$, mediated by every possible "pivot" node $j \in \mathcal{V}$.

For a given target pair $(i, k)$, its representation $\boldsymbol{R}'_{ik}$ is projected to form a **Query** vector, $\mathbf{q}_{ik}$. For each potential pivot $j$, the representations of the path segments, $\boldsymbol{R}'_{ij}$ and $\boldsymbol{R}'_{jk}$, are projected to form **Key** and **Value** vectors ($\mathbf{k}_{ij}, \mathbf{v}_{ij}$ and $\mathbf{k}_{jk}, \mathbf{v}_{jk}$). **Key** and **Value** vectors on the same path are combined by an operation $\mathcal{C}$ (e.g., addition or multiplication). Then, standard attention is applied:

$$\mathbf{k}_{ijk} = \mathcal{C}(\mathbf{k}_{ij}, \mathbf{k}_{jk}), \quad \mathbf{v}_{ijk} = \mathcal{C}(\mathbf{v}_{ij}, \mathbf{v}_{jk}) \tag{4}$$

$$\mathbf{o}_{ik} = \text{MultiheadAttention}_j(\mathbf{q}_{ik}, \mathbf{k}_{ijk}, \mathbf{v}_{ijk}) \tag{5}$$

For most graph tasks, an additive combine operation is suitable. The multiplicative operation may be useful for some geometric tasks (see Appendix B.1). The overall computational complexity of Pivotal Attention is $\mathcal{O}(N^3 \cdot d_r + N^2 \cdot d_r^2)$ (see Appendix B.4). A naive implementation of Pivotal Attention would create large intermediate tensors $\mathbf{k}_{ijk}$ and $\mathbf{v}_{ijk}$ of size $\mathcal{O}(N^3 \cdot d_r)$, which is memory-intensive for large graphs. To address this, we implemented a dedicated compute kernel that avoids storing these large intermediate tensors, reducing memory usage to $\mathcal{O}(N^2 \cdot d_r)$. While this optimization does not alter the theoretical computational cost, it improved training performance by more than 20 times in practice, as hardware memory bandwidth is a limiting factor.

## 2.4 PERMUTATION EQUIVARIANCE AND POSITIONAL ENCODINGS

FloydNet is inherently permutation equivariant and does not require Positional Encodings (PEs). Its core operation is fundamentally graph-structured. Permuting the node indices induces an equivalent permutation on the relationship tensor $\boldsymbol{R}$, and the symmetric aggregation over all pivots naturally preserves this equivariance.

## 2.5 GENERAL FORM OF FLOYDNET (k-FLOYDNET)

While a standard graph edge connects two vertices, a hypergraph edge can connect any number of vertices. The k-FloydNet is the general form of FloydNet that works on a hypergraph, updating hyper-edges $\vec{e} = (v_1, v_2, \ldots, v_k) \in V^{\times k}$ that connect to $k$ vertices, and uses a pivot $p \in V$ to enumerate all neighboring hyper-edges $\vec{e'}$ that differ from $\vec{e}$ by vertex $p$. The linear projection of embedding of hyper-edges $\vec{e'}$ generated from the same pivot $p$ are combined using operator $\mathcal{C}$ to form **Key** and **Value** vectors, which correspond to **Query** vectors projected from the embeddings of hyper-edges $\vec{e}$. We use notation $\vec{e}[i] \leftarrow p$ to represent the substitution of the $i$-th vertex in hyper-edge $\vec{e}$ with the pivot $p$, thereby constructing a neighboring hyper-edge $\vec{e'} = (v_1, v_2, \ldots, v_{i-1}, p, v_{i+1}, \ldots, v_{k-1}, v_k)$. The formula of Pivotal Attention of k-FloydNet is:

$$\mathbf{k}_{\vec{e},p} = \mathcal{C}(\mathbf{k}_{\vec{e}[1]\leftarrow p}, \mathbf{k}_{\vec{e}[2]\leftarrow p}, \ldots, \mathbf{k}_{\vec{e}[k]\leftarrow p}) \tag{6}$$

$$\mathbf{v}_{\vec{e},p} = \mathcal{C}(\mathbf{v}_{\vec{e}[1]\leftarrow p}, \mathbf{v}_{\vec{e}[2]\leftarrow p}, \ldots, \mathbf{v}_{\vec{e}[k]\leftarrow p}) \tag{7}$$

$$\mathbf{o}_{\vec{e}} = \text{MultiheadAttention}_p(\mathbf{q}_{\vec{e}}, \mathbf{k}_{\vec{e},p}, \mathbf{v}_{\vec{e},p}) \tag{8}$$

**FloydNet is the special case where k = 2, and self-attention of the Transformer is the special case where k=1**. The expressive power of k-FloydNet increases with the value of $k$, as higher-order interactions among sets of vertices can be captured more effectively. The overall computational complexity of Pivotal Attention is $\mathcal{O}(N^{k+1} \cdot d_r + N^k \cdot d_r^2)$ (see Appendix B.3 and Appendix B.4).

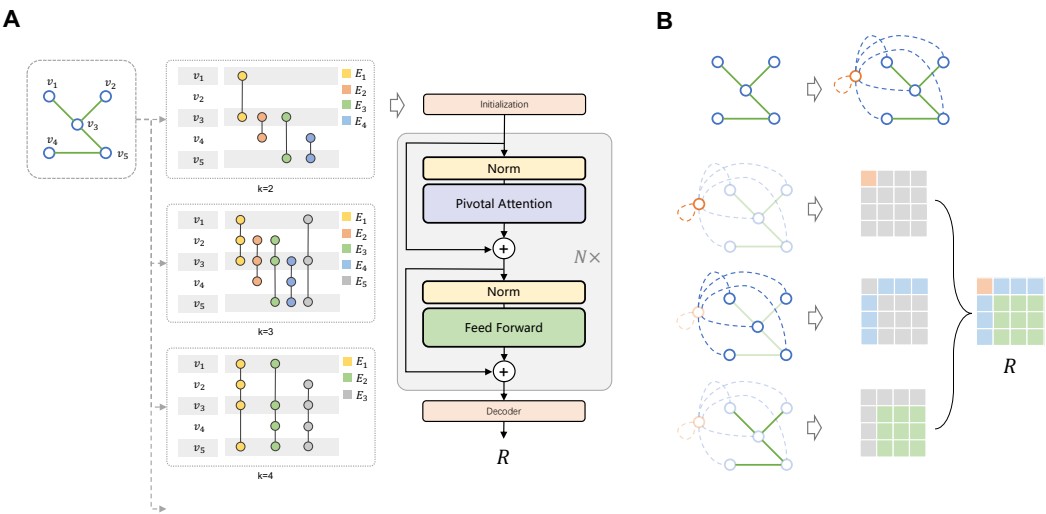

Figure 2: **The FloydNet framework. (A)** The model transforms various graph inputs into a unified relational representation $\boldsymbol{R}$. **(B)** The SuperNode provides a unified representation for graph, node, and edge information within $\boldsymbol{R}$.

## 3 THEORETICAL ANALYSIS OF k-FLOYDNET

We now formally analyze the expressive power of k-FloydNet. We leverage the connection between our architecture and the Weisfeiler-Lehman (WL) hierarchy.

**Theorem 1** (k-FloydNet is equivalent to k-FWL)**.** *The k-FloydNet architecture functionally implements the k-Folklore Weisfeiler-Lehman (k-FWL) color refinement algorithm. Consequently, it can distinguish any pair of non-isomorphic graphs distinguishable by the k-FWL test.*

*Proof Sketch.* The k-FWL test refines the color of a k-tuple based on the multiset of colors of its neighbors (formed by swapping one node with a pivot). Pivotal Attention (Section 2) performs exactly this operation in a continuous embedding space: it aggregates information from all neighbor tuples generated by a pivot $p$. Since the attention mechanism is permutation-invariant with respect to $p$ and can model arbitrary non-linear hashing functions (via MLPs), it satisfies the conditions of the k-FWL update rule. (See Appendix C for full proof). $\square$

**Corollary (3-WL Expressiveness):** Since FloydNet corresponds to 2-FloydNet, and it is established that 2-FWL is equivalent to 3-WL (Cai et al., 1992), FloydNet achieves **3-WL expressive power**. This strictly surpasses the 1-WL limit of standard MPNNs.

**Theorem 2** (Principled Long-Range Information Propagation)**.** *After $L$ layers, the representation $\boldsymbol{R}_{ik}^{(L)}$ in FloydNet integrates information from all relational paths of length up to $2^L$ between nodes $i$ and $k$.*

*Proof Sketch.* By induction. For $L = 1$, $\boldsymbol{R}_{ik}$ aggregates paths $i \rightsquigarrow j \rightsquigarrow k$ (length 2). If layer $L - 1$ captures length $2^{L-1}$, then combining two such segments via a pivot concatenates the paths, covering length $2^{L-1} + 2^{L-1} = 2^L$. This exponential growth in receptive field mitigates the over-squashing problem common in MPNNs. $\square$

## 4 EXPERIMENTS

To validate the design of FloydNet, we conduct a comprehensive set of experiments designed to answer three key questions: (1) Does it possess the theoretical expressive power to overcome the limitations of current GNNs? (2) Can it learn complex, global algorithmic reasoning procedures? (3) Does this capability translate to strong performance on challenging real-world problems? We address these questions by evaluating our model on graph property prediction, neural algorithmic reasoning, Traveling Salesman Problem and LRGB.

Our code, data generation scripts, and exact hyperparameters are available at `http://anonymous_for_submission`. Where applicable (e.g., CLRS), we use the official evaluation harness without hints at test time unless explicitly stated; for TSP we guarantee train/test isolation across sizes and random seeds and retain only instances with unique optima to avoid solution multiplicity confounds.

### 4.1 EXPRESSIVE POWER: GRAPH PROPERTY PREDICTION

We first measure FloydNet's ability to overcome the limitations of standard GNNs on two complementary fronts: quantitative structure counting and qualitative isomorphism testing.

**Quantitative Expressiveness on Homomorphism Counting Tasks.** To probe the model's ability to count complex graph structures, we use the synthetic property prediction tasks from Zhang et al. (2024a). These benchmarks, which involve homomorphism and subgraph isomorphism counting, act as a practical barometer for transcending the 1-WL test. As shown in Tables 1 and 3, while standard MPNNs fail on most tasks, FloydNet consistently reduces the Mean Absolute Error to near-zero across all graph, node, and edge-level tasks. This strong quantitative performance provides clear initial evidence that FloydNet operates well beyond the 1-WL limit.

Table 1: MAE on homomorphism counting. FloydNet encodes all substructures and achieves near-zero error, substantially outperforming all baselines.

| Task / Model | Graph-level | | | Node-level | | Edge-level | | |
|---|---|---|---|---|---|---|---|---|
| MPNN (GIN) | .300 | .233 | .254 | .505 | .478 | - | - | - |
| Subgraph GNN | .011 | .015 | .012 | .004 | .058 | .003 | .058 | .048 |
| Local 2-GNN | .008 | .008 | .010 | .003 | .004 | .005 | .006 | .008 |
| Local 2-FGNN | .003 | .005 | .004 | .005 | .005 | .007 | .007 | .008 |
| **FloydNet** | .000 | .001 | .001 | .000 | .000 | .000 | .000 | .000 |

Table 2: Accuracy on the BREC benchmark. k-FloydNet's distinguishing power is empirically identical to the k-FWL test.

| Model / Heuristic | Accuracy (%) |
|---|---|
| MPNN (1-WL equiv.) | <20% |
| Graphormer | 19.8% |
| PPGT | 58.5% |
| KP-GNN | 68.8% |
| 3-WL(2-FWL) Test | **67.5%** |
| **FloydNet** | **67.5%** |
| **3-FloydNet** | **95.0%** |
| **4-FloydNet** | **99.8%** |

Table 3: MAE on the (Chordal) Cycle Counting task. FloydNet accurately counts all types of cycles and substantially outperforms all baselines.

| Task / Model | Graph-level | | | | | | Node-level | | | | | | Edge-level | | | | | |
|---|---|---|---|---|---|---|---|---|---|---|---|---|---|---|---|---|---|---|
| MPNN | .358 | .208 | .188 | .146 | .261 | .205 | .600 | .413 | .300 | .207 | .318 | .237 | - | - | - | - | - | - |
| Subgraph GNN | .010 | .020 | .024 | .046 | .007 | .027 | .003 | .005 | .092 | .082 | .050 | .073 | .001 | .003 | .090 | .096 | .038 | .065 |
| Local 2-GNN | .008 | .011 | .017 | .034 | .007 | .016 | .002 | .005 | .010 | .023 | .004 | .015 | .001 | .005 | .010 | .019 | .005 | .014 |
| Local 2-FGNN | .003 | .004 | .010 | .020 | .003 | .010 | .004 | .006 | .012 | .021 | .004 | .014 | .003 | .006 | .012 | .022 | .005 | .012 |
| **FloydNet** | .000 | .000 | .001 | .001 | .000 | .001 | .000 | .000 | .000 | .000 | .000 | .000 | .000 | .000 | .000 | .000 | .000 | .000 |

**Qualitative Expressiveness on Isomorphism Testing.** To precisely characterize FloydNet's power in distinguishing non-isomorphic graphs, we evaluate it on the challenging BREC benchmark (Wang & Zhang, 2024). The results, summarized in Table 2, lead to a key finding: FloydNet's performance is identical to the theoretical 3-WL heuristic.

**Expressiveness of k-FloydNet.** We further empirically validate our theoretical claims regarding k-FloydNet. The results, summarized in Table 2 and detailed in Appendix D.1, confirm that the architecture demonstrates scalable expressive power consistent with the theory. As shown in Table 2, 3-FloydNet and 4-FloydNet achieve accuracies of 95.0% and 99.8% respectively on BREC. These results align perfectly with our theoretical analysis in Section 3, confirming that the practical distinguishing power of the k-FloydNet model matches the theoretical k-FWL hierarchy.

## 4.2 NEURAL ALGORITHMIC REASONING

We now evaluate FloydNet's core capability: learning algorithmic computations.

**The CLRS Benchmark.** The CLRS-30 benchmark (Veličković et al., 2022) is a standard suite for assessing algorithmic reasoning. We follow the standard CLRS-30 protocol, training FloydNet model for 30 algorithmic tasks. Evaluation is performed on out-of-distribution instances with larger problem sizes, without providing any ground-truth hints to the model at test time. We evaluate on the full benchmark and report aggregated performance across eight categories in Table 4. The results show that FloydNet consistently and significantly outperforms previous state-of-the-art models. Notably, it achieves near-perfect scores ($> 95\%$) across most of the algorithms, including those that are difficult for standard GNNs , such as Strings, Graph algorithms (e.g., Floyd-Warshall), and Sorting. We also found that intermediate result (hints) are not as beneficial as expected: for most algorithms, using hints can actually degrade performance. To rigorously test length scalability, we extended the evaluation protocol to problem sizes up to $N = 256$, following the scale of the SALSA-CLRS benchmark (Minder et al., 2023). As visualized in Figure 3(a), FloydNet maintains robust performance at this scale, with the no-hint model showing superior extrapolation capabilities. Full per-algorithm results and detailed extrapolation scores are available in Appendix D.2.

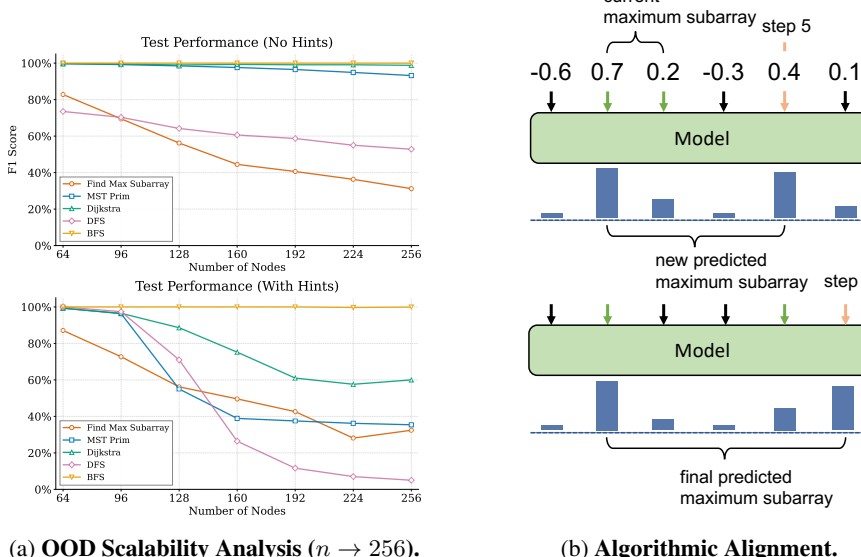

(a) **OOD Scalability Analysis ($n \to 256$).**  (b) **Algorithmic Alignment.**

Figure 3: **Probing the Scalability and Reasoning Logic of FloydNet on CLRS. (a)** We extend the OOD evaluation to problem sizes up to $n = 256$, following the protocol of Minder et al. (2023). The results indicate that models trained *without hints* (upper) generally exhibit superior length extrapolation capabilities compared to those trained *with hints* (lower). **(b)** Visualization of the internal reasoning process for the Maximum Subarray problem (Kadane's algorithm). The model iteratively predicts the new maximum subarray based on the current maximum subarray and the next number corresponding to the current step. The clear alignment demonstrates that FloydNet has successfully learned to emulate the algorithm's state-update logic.

**Probing Algorithmic Reasoning.** To verify that FloydNet learns the underlying algorithmic process, we probed its step-by-step reasoning on dynamic programming tasks. We use *Find Max. Subarray (Kadane's algorithm)* as a case study, where the correct solution requires maintaining and updating the optimal subarray at each step of a traversal. As visualized in Figure 3b, FloydNet's strong alignment with intermediate trajectory demonstrates that FloydNet is not merely memorizing input-output pairs but has successfully learned to emulate the core, state-updating logic of the algorithm.

Table 4: Average test accuracy (%) on the CLRS-30 benchmark, aggregated by algorithm category. $N$ denotes the number of algorithms in each group. All models are trained on problem sizes of $n = 16$ and tested on larger instances of $n = 64$.

| Class | Model | Sort (N=4) | Search (N=3) | D&C (N=1) | Greedy (N=2) | DP (N=3) | Graph (N=12) | String (N=2) | Geometry (N=3) | Total (N=30) |
|---|---|---|---|---|---|---|---|---|---|---|
| **GNN** | **Triplet-GMPNN** (Ibarz et al., 2022) | 75.64 | 58.83 | 76.36 | 91.22 | 81.99 | 86.42 | 49.09 | 94.09 | 80.04 |
| | **RANR** (Xu & Veličković, 2024) | 94.15 | 82.90 | 83.53 | 91.66 | 42.69 | 74.15 | 49.12 | 88.44 | 75.78 |
| | **G-ForgetNet** (Bohde et al., 2024) | 78.09 | 63.84 | 78.97 | 91.79 | 86.70 | 88.80 | 54.73 | 95.09 | 82.89 |
| **GT** | **RT** (Diao & Loynd, 2022) | 50.01 | 65.31 | 66.52 | 85.33 | 83.20 | 65.33 | 32.52 | 84.55 | 66.18 |
| | **ET** (Müller et al., 2024) | 82.26 | 63.00 | 64.44 | 81.67 | 83.49 | 86.08 | 54.84 | 88.22 | 80.13 |
| **Ours** | **FloydNet** | 100.00 | 91.60 | 86.20 | 93.17 | 90.02 | 98.60 | 99.72 | 99.68 | 96.64 |

### 4.3 PUSHING THE LIMITS: THE TRAVELING SALESMAN PROBLEM (TSP).

To test our model on a canonical NP-hard problem, we evaluate it on the Traveling Salesman Problem (TSP). The difficulty stems from the factorial growth in potential solutions, making the search for the shortest tour computationally intractable for all but the smallest instances.

**Settings and Training.** We constructed two new benchmarks for this task. The non-metric TSP dataset is composed of random complete graphs with up to 200 nodes, where edge weights are integers uniformly sampled from $[1, 100]$. The metric TSP dataset is generated by uniformly sampling $N$ unique integer coordinates $(X, Y)$ from the range $[1, 100]$ in 2D space, with edge weights defined as the Euclidean distances between coordinates. Since standard TSP instances often possess multiple optimal solutions, training a deterministic regressor can lead to mode averaging. To address this, we formulate the training within a Denoising Diffusion Probabilistic Model (DDPM) framework(Ho et al., 2020). This allows FloydNet to learn the underlying distribution of optimal tours, naturally handling instances with multiple valid solutions. To ensure a fair and unambiguous evaluation, we use the state-of-the-art exact solver, Concorde (Applegate et al., 1998; Aldous & Percus, 2003), to find the ground-truth optimal tour. Our model is trained exclusively on smaller graphs ($N \leq 100$) and tested on larger, unseen graphs ($100 < N \leq 200$) to assess its out-of-distribution generalization. We benchmark FloydNet against Linkern, a highly-optimized and widely-used heuristic solver based on the Chained Lin-Kernighan algorithm (Lin & Kernighan, 1973). To further analyze the capability of FloydNet, we constructed subsets of both the Metric and Non-Metric datasets by retaining only those instances possessing a unique optimum. Subsequently, we evaluate our model on these single-solution datasets. A detailed description of the problem, solvers, and our data generation pipeline is provided in Appendix D.3.

**Result.** On held-out general TSP instances with $100 < N \leq 200$, our model demonstrates a strong ability to significantly improve the optimality rate through multiple sampling as shown in Figure 4a. Specifically, it increases the optimality rate from $83.6\%$ to $99.8\%$ after 10 samples, substantially surpassing the $38.8\%$ achieved by the strong Linkern heuristic. Furthermore, it exhibits excellent extrapolation capabilities concerning the number of nodes ($N$). As $N$ increases, the optimality rate of Linkern rapidly deteriorates, dropping to $16.1\%$ for instances where $180 \leq N \leq 200$. In sharp contrast, our model with 10 samples maintains a robust optimality rate of $99.4\%$ in this challenging range. We also examined a subset containing only single-solution instances. On non metric TSP instances with $100 < N \leq 200$, the FloydNet finds the exact optimal tour in $92.6\%$ of cases, as shown in Figure 4b. This performance markedly outperforms the Linkern heuristic, which achieves only a $15.7\%$ optimality rate on these larger instances. The performance gap widens significantly with instance size: Linkern's optimality rate plummets to $1.3\%$ when $180 \leq N \leq 200$, while FloydNet still achieves $88.3\%$. For smaller validation instances ($N \leq 100$), FloydNet demonstrates robust performance within the training distribution, with an optimality rate exceeding $96\%$. On Metric TSP instances, the Linkern heuristic performs substantially better than on Non-Metric TSP instances, leveraging the underlying geometric properties. In this setting, FloydNet achieves an optimality rate that is comparable to that of the Linkern heuristic.

**Constraint and error analysis.** We analyzed the failure cases of FloydNet on singe-solution instances and found that only a small fraction ($0.8\%$) corresponded to valid but suboptimal TSP tours, $6.2\%$ produced multiple disjoint cycles covering all nodes, and the remaining $93\%$ generated fewer edges than nodes, resulting in incomplete tours that failed to visit all nodes. We hypothesize that this phenomenon may stem from numerical issues or interactions with mechanisms such as initialization, normalization, or optimization. Notably, scaling up the model effectively mitigates the effect, highlighting a promising direction for future study.

**Scaling Properties.** As shown in Figures 4d and 4e, performance consistently improves with both larger training sets and increased model capacity. To substantiate this observation, we conducted two sets of controlled experiments with FloydNet on single-solution instances. In the first, we fixed the training set size and varied the model capacity from $L = 1$ to $L = 96$ layers, with larger values infeasible due to memory constraints; we also fixed the model capacity and varied the number of training samples. In both cases, we evaluated performance on out-of-distribution test instances to assess generalization. All experiments were trained under identical hyperparameter settings and terminated after a total of 2.5M training samples. The results demonstrate that FloydNet benefits from both data and model size scaling, in contrast to many traditional MPNNs whose performance typically deteriorates with depth due to over-smoothing. Thanks to its global refinement mechanism, FloydNet avoids this pitfall and establishes itself as a scalable architecture, where increasing model size and data directly translates into stronger reasoning capabilities.

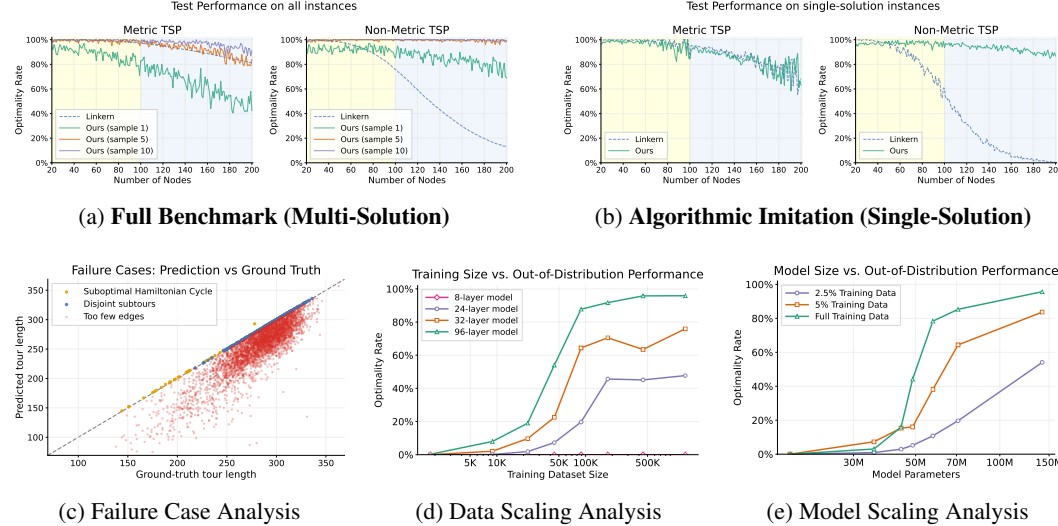

(a) **Full Benchmark (Multi-Solution)**     (b) **Algorithmic Imitation (Single-Solution)**

(c) Failure Case Analysis     (d) Data Scaling Analysis     (e) Model Scaling Analysis

Figure 4: **Analysis of FloydNet on the TSP.** **(a)** Optimality rate on the benchmark (including multi-solution instances). The **yellow region** denotes validation on in-distribution sizes ($N \leq 100$), while the **cyan region** demonstrates OOD generalization to larger problem sizes ($100 < N \leq 200$). **(b)** Performance on the subset of single-solution instances, testing the model's precise alignment with a unique optimal ground truth. **(c)** Distribution of error types. **(d-e)** Scaling laws showing that FloydNet effectively utilizes increased data and model depth, avoiding the over-smoothing issues typical of deep MPNNs.

## 4.4 REAL-WORLD APPLICATIONS

Finally, we validate FloydNet's practical utility on real-world benchmarks where long-range interactions are critical. The results on the Long-Range Graph Benchmark (LRGB) (Dwivedi et al., 2022), shown in Table 5, highlight the strengths of our model's relational reasoning paradigm. Without tuning, FloydNet establishes a new state-of-the-art by a significant margin on the link prediction task (PCQM-Contact) and also secures top performance on the node classification task COCO-SP and ZINC.

On the remaining graph-level and node-level tasks, FloydNet remains highly competitive with other leading models. Our analysis indicates that performance on these tasks, particularly graph property prediction, is constrained by overfitting on the training set. A detailed discussion of this phenomenon is provided in Appendix D.4.

Table 5: Mean performance on Long-Range Graph Benchmarks over five runs.

| Class | Task
Dataset
Metric | Graph Class.
Peptides-Func
AP ↑ | Graph Reg.
Peptides-Struct
MAE ↓ | Node Class.
PascalVOC-SP
F1 ↑ | Node Class.
COCO-SP
F1 ↑ | Link Pred.
PCQM-Contact
MRR ↑ |
|---|---|---|---|---|---|---|
| **GNN** | GCN(Brossard et al., 2020) | 0.5930 | 0.3496 | 0.1268 | 0.0841 | 0.3234 |
| **GT** | Exphormer(Shirzad et al., 2023) | 0.6258 | 0.2512 | 0.3446 | 0.3430 | 0.3587 |
|  | GPS(Rampášek et al., 2022) | 0.6535 | 0.2500 | 0.3724 | 0.3774 | 0.3437 |
| **SSM** | Graph-Mamba(Wang et al., 2024) | 0.6739 | 0.2478 | 0.4191 | 0.3960 | 0.3395 |
| **Ours** | **FloydNet** | **0.6224** | **0.2612** | **0.4046** | **0.4901** | **0.6143** |

## 5 RELATED WORK

### 5.1 ARCHITECTURES BEYOND 1-WL EXPRESSIVENESS

A central theme in GNN research is overcoming the expressive limitations of MPNNs, famously bounded by the 1-WL test (Xu et al., 2019; Morris et al., 2019). This pursuit has led to a variety of more powerful architectures, including Subgraph GNNs (Bevilacqua et al., 2022; Zhang & Li, 2021), Spectral GNNs(Zhang et al., 2024b; Lim et al., 2023) and higher-order GNNs(Maron et al., 2019b;a) grounded in more powerful theoretical tests. Among these, the k-Folklore Weisfeiler-Lehman (k-FWL) test (Cai et al., 1992; Maron et al., 2019a) serves as a key benchmark for models that reason over tuples of nodes. The 2-FWL test, for instance, refines representations for pairs of nodes $(u, v)$ by aggregating over all intermediate pivot nodes $w$, considering relational triangles of the form $u \rightsquigarrow w \rightsquigarrow v$. Architectures like the Folklore GNN (FGNN) (Zhang et al., 2024a) are designed to mimic this process.

While many expressive models rely on enriching local structures or implementing complex tensor operations, FloydNet introduces a distinct computational paradigm. It is based on learning a global, DP-style operator that is algorithmically aligned with the pivotal aggregation pattern of the FWL tests. Our empirical results strongly support a key theoretical connection: the relational refinement in k-FloydNet is not merely a novel heuristic but is empirically equivalent in expressive power to the k-FWL test. We give a formal discussion on Appendix C. This result precisely characterizes k-FloydNet, positioning it as an elegant and practical realization of this powerful theoretical class.

### 5.2 TRANSFORMERS AND POSITIONAL ENCODINGS

The standard Transformer architecture (Vaswani et al., 2017) excels at processing 1D sequences. However, graphs lack a canonical ordering, making the direct application of Transformers challenging. Graph Transformers (GTs) (Dwivedi & Bresson, 2021; Ying et al., 2021; Kim et al., 2022; Rampasek et al., 2022; Lai et al., 2023) adapt this architecture by treating nodes as an unordered set, but this makes them inherently blind to the graph's topology. To compensate, they require carefully designed Positional and Structural Encodings (PSEs) to re-inject the structural information that was lost.

This work proposes a more principled generalization. Instead of operating on a 1D set of node representations, FloydNet's Pivotal Attention mechanism operates directly on the 2D relational space of all node pairs. This perspective sidesteps the need for PSEs entirely. While many PSEs have been developed, often based on Laplacian eigenvectors (Kreuzer et al., 2021), they introduce their own challenges, such as eigenvector ambiguity (Lim et al., 2023). Even more robust spectral features (Huang et al., 2024; Li et al., 2020) have fundamental theoretical limits; a comprehensive analysis by Zhang et al. (2024b) shows their expressive power is strictly weaker than the 3-WL test. Thus, most GTs, despite their architectural complexity, operate below the expressiveness ceiling that FloydNet empirically achieves. By building the graph's relational structure into its core computational block, FloydNet provides an elegant and powerful alternative that is naturally permutation-equivariant.

## 6 CONCLUSION AND FUTURE WORK

This paper introduced FloydNet, a new paradigm for graph reasoning based on a learned, DP-style global refinement operator that replaces local message passing. Theoretically, FloydNet and its' extends (k-FloydNet) is equivalent to the powerful 2-FWL(k-FWL) test, yet empirically, it achieves such distinguishing power on challenging benchmarks. This expressiveness translates to state-of-the-art performance on algorithmic reasoning (CLRS-30) and combinatorial optimization (TSP), establishing our approach as a powerful and practical paradigm for global graph reasoning.

Future work can focus on improving scalability beyond the current $\mathcal{O}(N^3)$ complexity through sparse mechanisms, extending the framework to new domains like multimodality for which its relational structure is a natural fit, and further strengthening its theoretical foundations by connecting to Abstract DP theory(Bertsekas, 2022; 2025). Ultimately, we envision FloydNet as a core component of future System-2 reasoning systems.

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

# Appendix

## Table of Contents

## A EXPERIMENTAL SETUP AND IMPLEMENTATION DETAILS

### A.1 GENERAL IMPLEMENTATION AND MODEL CONFIGURATIONS

Across all experiments, FloydNet models were implemented in either PyTorch(Paszke et al., 2019) or JAX(Bradbury et al., 2018). The core architecture consists of a stack of $L$ `FloydBlocks`. Each block uses a multi-head Pivotal Attention layer followed by a standard MLP-based Feed-Forward Network (FFN). The FFN consists of two linear layers with a GELU activation in between. We use pre-layer normalization (LayerNorm) throughout the network.

We detail the architecture and training configurations for each benchmark dataset. Unless otherwise specified, we use a batch size of 1 for simplicity and set the AdamW momentum parameters to [0.9, 0.95].

**NED Counting.** For the NED counting dataset we use a model with $L = 32$ layers, hidden representation dimension $d_r = 128$, and 2 attention heads. Batch size = 64. Models are optimized with AdamW using an initial learning rate of $1 \times 10^{-4}$ and a Reduce-on-Plateau learning-rate scheduler. The training objective is normalized mean absolute error (MAE) to match the baseline; early stopping is applied based on validation MAE.

**BREC.** For the BREC dataset we use $L = 32$ layers, $d_r = 2$, and a single attention head. We replace layer norm with batch normalization, remove the feed-forward network (FFN) inside the pivotal-attention block, and perform all computations in `float64` precision.

**CLRS.** For the CLRS benchmark we follow the official framework: FloydNet is implemented as a processor while preserving the original encoder, decoder, and loss functions. For each algorithm, we select the deepest model that fits within our GPU memory constraints (layer counts vary with the algorithm's hint size). Training uses AdamW with an initial learning rate of $1 \times 10^{-4}$, a short linear warmup, and cosine decay thereafter. Each algorithm is trained for up to 80k steps and evaluation is performed using the best validation checkpoint. Batch size, weight decay, and other hyperparameters follow the CLRS framework's default setting.

**TSP.** For the traveling salesman (TSP) experiments we use a larger model with $d_r = 384$ and 6 attention heads. We train multiple models with layer depths ranging from $L = 8$ to $L = 96$ to study model scaling effects. The loss is binary cross-entropy applied to edge predictions. Optimization uses AdamW with a fixed learning rate $1 \times 10^{-4}$ together with a Reduce-on-Plateau scheduler. Training is conducted in data-parallel mode across 64 GPUs, for 400 epochs, each consisting of 100 steps.

**Real-world tasks.** For real-world applications LRGB and ZINC, we use $L = 48$, $d_r = 384$, and 6 attention heads. Optimization follows the same recipe: AdamW with initial learning rate $1 \times 10^{-4}$ and a Reduce-on-Plateau scheduler.

### A.2 DATASET DETAILS

**NED Counting.** This synthetic benchmark from Zhang et al. (2024a) is designed to quantitatively assess the expressive power of GNNs beyond the qualitative WL test. The core task is homomorphism counting, where models predict the frequencies of eight specific pattern graphs (e.g., chordal cycles) within larger target graphs. Each of these patterns corresponds to a well-defined level of structural complexity, captured by its Nested Ear Decomposition (NED) type. Thus, a model's quantitative accuracy on these tasks directly reflects its ability to detect and reason about complex structures. For a comprehensive evaluation, the dataset provides tasks at three levels: graph-level (testing invariance), node-level, and edge-level (testing equivariance).

We observed that FloydNet tends to drive the training loss to zero prematurely on the graph-level tasks , which in turn hampers generalization to the validation and test sets. To mitigate this, and following the original dataset generation protocol of Chen et al. (2020), we regenerated a larger training set consisting of 50K randomly sampled graphs, which we used exclusively for the graph-level tasks. This adjustment stabilizes training and yields more reliable evaluation results, reducing the test error from 0.005–0.01 to nearly zero.

**BREC.** The Benchmark for Realized GNN Expressiveness (BREC), introduced by Wang & Zhang (2024), is a dataset designed to address the shortcomings of previous benchmarks in evaluating

GNN expressiveness. It provides a more challenging, fine-grained, and larger-scale testbed than prior datasets (e.g., EXP, CSL). Its goal is to empirically measure the realized expressiveness of GNNs—the practical distinguishing power an implemented model can achieve. Which allows for a much finer-grained comparison between models that are all simply "more powerful than 1-WL."

- The dataset contains 800 unique non-isomorphic graphs organized into 400 challenging pairs. These graphs are grouped into 4 distinct categories, each with a specific quantitative breakdown:

  (i) **Basic**: 60 pairs of simple, non-regular, 1-WL-indistinguishable graphs.
  (ii) **Regular**: 140 pairs, further subdivided into four types of regular graphs with increasing complexity (50 simple regular, 50 strongly regular, 20 4-vertex condition, and 20 distance regular pairs).
  (iii) **Extension**: 100 pairs designed to test specific theoretical GNN extensions (e.g., substructure-counting vs. k-hop subgraphs).
  (iv) **CFI**: 100 pairs generated using the Cai-Furer-Immerman construction, with their difficulty precisely calibrated against the WL hierarchy (specifically, 60 pairs distinguishable by 3-WL, 20 pairs by 4-WL, and 20 pairs that remain 4-WL-indistinguishable).

**CLRS-30.** The CLRS-30 benchmark (Veličković et al., 2022) is a comprehensive suite of algorithmic reasoning tasks, serving as a standardised testbed for evaluating a model's ability to learn classical algorithms. It is designed to directly assess a model's capacity for complex reasoning and out-of-distribution (OOD) generalization, where OOD specifically refers to executing a learned algorithm on inputs of a larger size (e.g., training on arrays of size 16 and testing on size 64). The benchmark is quantitatively composed of 30 classical algorithmsrom the textbook introduction to Algorithms(Cormen et al., 2022), which are grouped into 8 distinct categories such as sorting, searching, and dynamic programming. The core logical format of the dataset represents every algorithm's execution as a trajectory on a graph. Each data sample contains the problem's initial inputs, the final correct outputs, and a time-series of intermediate states called hints. Crucially, all these components are mapped to a unified graph representation as node, edge, or global graph features. This standardized format is the key that enables a single GNN processor to be trained across multiple, structurally diverse algorithms, a central goal of our work.

# B FloydNet Architectural Details and Extensions

## B.1 Multiplicative Pivotal Attention

The main paper focuses on the additive combine operation for keys and values ($k_{ik,j} = k_{ij} + k_{jk}$) due to its computational efficiency. However, a multiplicative combine operation, such as element-wise multiplication ($k_{ik,j} = k_{ij} \odot k_{jk}$), offers greater expressive power for certain tasks. When pairwise features parameterize geometric transforms (e.g., SE(3) transformations), a multiplicative combine naturally models the composition of transformations along a path $i \rightsquigarrow j \rightsquigarrow k$. For instance, if $\mathbf{T}_{ij}$ represents a $3 \times 3$ rotation matrix, the composition along the path is $\mathbf{T}_{ik} = \mathbf{T}_{jk} \cdot \mathbf{T}_{ij}$.

We prove the multiply version can express composition of rotation matrices. We embed the elements of rotation matrices $T_A, T_B$ into relationship tensor $R_A = (T_{A,1,1}, T_{A,1,2}, \ldots, T_{A,3,2}, T_{A,3,3})$ and $R_B = (T_{B,1,1}, T_{B,1,2}, \ldots, T_{B,3,2}, T_{B,3,3})$. According to the architecture of FloydNet, the relationship tensor will be linear transformed to get $V_A, V_B$, and then element-wise multiplied to get $V_C$, and then linear projected to get attention output $O$. The transformation matrices for $V_A, V_B$ and are $M_A, M_B$, and the linear projection matrix is $M_C$. Assuming the values of $M_A, M_B, M_C$ are:

$$
M_A = \begin{pmatrix}
1 & 1 & 1 & 0 & 0 & 0 & 0 & 0 & 0 & 0 & 0 & 0 & 0 & 0 & 0 & 0 & 0 & 0 & 0 & 0 & 0 & 0 & 0 & 0 & 0 & 0 & 0 \\
0 & 0 & 0 & 1 & 1 & 1 & 0 & 0 & 0 & 0 & 0 & 0 & 0 & 0 & 0 & 0 & 0 & 0 & 0 & 0 & 0 & 0 & 0 & 0 & 0 & 0 & 0 \\
0 & 0 & 0 & 0 & 0 & 0 & 1 & 1 & 1 & 0 & 0 & 0 & 0 & 0 & 0 & 0 & 0 & 0 & 0 & 0 & 0 & 0 & 0 & 0 & 0 & 0 & 0 \\
0 & 0 & 0 & 0 & 0 & 0 & 0 & 0 & 0 & 1 & 1 & 1 & 0 & 0 & 0 & 0 & 0 & 0 & 0 & 0 & 0 & 0 & 0 & 0 & 0 & 0 & 0 \\
0 & 0 & 0 & 0 & 0 & 0 & 0 & 0 & 0 & 0 & 0 & 0 & 1 & 1 & 1 & 0 & 0 & 0 & 0 & 0 & 0 & 0 & 0 & 0 & 0 & 0 & 0 \\
0 & 0 & 0 & 0 & 0 & 0 & 0 & 0 & 0 & 0 & 0 & 0 & 0 & 0 & 0 & 1 & 1 & 1 & 0 & 0 & 0 & 0 & 0 & 0 & 0 & 0 & 0 \\
0 & 0 & 0 & 0 & 0 & 0 & 0 & 0 & 0 & 0 & 0 & 0 & 0 & 0 & 0 & 0 & 0 & 0 & 1 & 1 & 1 & 0 & 0 & 0 & 0 & 0 & 0 \\
0 & 0 & 0 & 0 & 0 & 0 & 0 & 0 & 0 & 0 & 0 & 0 & 0 & 0 & 0 & 0 & 0 & 0 & 0 & 0 & 0 & 1 & 1 & 1 & 0 & 0 & 0 \\
0 & 0 & 0 & 0 & 0 & 0 & 0 & 0 & 0 & 0 & 0 & 0 & 0 & 0 & 0 & 0 & 0 & 0 & 0 & 0 & 0 & 0 & 0 & 0 & 1 & 1 & 1
\end{pmatrix}
$$

$$M_B = \begin{pmatrix} 1 & 0 & 0 & 0 & 0 & 0 & 0 & 0 & 0 & 1 & 0 & 0 & 0 & 0 & 0 & 0 & 0 & 0 & 1 & 0 & 0 & 0 & 0 & 0 & 0 & 0 & 0 \\ 0 & 1 & 0 & 0 & 0 & 0 & 0 & 0 & 0 & 0 & 1 & 0 & 0 & 0 & 0 & 0 & 0 & 0 & 0 & 1 & 0 & 0 & 0 & 0 & 0 & 0 & 0 \\ 0 & 0 & 1 & 0 & 0 & 0 & 0 & 0 & 0 & 0 & 0 & 1 & 0 & 0 & 0 & 0 & 0 & 0 & 0 & 0 & 1 & 0 & 0 & 0 & 0 & 0 & 0 \\ 0 & 0 & 0 & 1 & 0 & 0 & 0 & 0 & 0 & 0 & 0 & 0 & 1 & 0 & 0 & 0 & 0 & 0 & 0 & 0 & 0 & 1 & 0 & 0 & 0 & 0 & 0 \\ 0 & 0 & 0 & 0 & 1 & 0 & 0 & 0 & 0 & 0 & 0 & 0 & 0 & 1 & 0 & 0 & 0 & 0 & 0 & 0 & 0 & 0 & 1 & 0 & 0 & 0 & 0 \\ 0 & 0 & 0 & 0 & 0 & 1 & 0 & 0 & 0 & 0 & 0 & 0 & 0 & 0 & 1 & 0 & 0 & 0 & 0 & 0 & 0 & 0 & 0 & 1 & 0 & 0 & 0 \\ 0 & 0 & 0 & 0 & 0 & 0 & 1 & 0 & 0 & 0 & 0 & 0 & 0 & 0 & 0 & 1 & 0 & 0 & 0 & 0 & 0 & 0 & 0 & 0 & 1 & 0 & 0 \\ 0 & 0 & 0 & 0 & 0 & 0 & 0 & 1 & 0 & 0 & 0 & 0 & 0 & 0 & 0 & 0 & 1 & 0 & 0 & 0 & 0 & 0 & 0 & 0 & 0 & 1 & 0 \\ 0 & 0 & 0 & 0 & 0 & 0 & 0 & 0 & 1 & 0 & 0 & 0 & 0 & 0 & 0 & 0 & 0 & 1 & 0 & 0 & 0 & 0 & 0 & 0 & 0 & 0 & 1 \end{pmatrix}$$

$$M_C = \begin{pmatrix} 1 & 0 & 0 & 1 & 0 & 0 & 1 & 0 & 0 & 0 & 0 & 0 & 0 & 0 & 0 & 0 & 0 & 0 & 0 & 0 & 0 & 0 & 0 & 0 & 0 & 0 & 0 \\ 0 & 1 & 0 & 0 & 1 & 0 & 0 & 1 & 0 & 0 & 0 & 0 & 0 & 0 & 0 & 0 & 0 & 0 & 0 & 0 & 0 & 0 & 0 & 0 & 0 & 0 & 0 \\ 0 & 0 & 1 & 0 & 0 & 1 & 0 & 0 & 1 & 0 & 0 & 0 & 0 & 0 & 0 & 0 & 0 & 0 & 0 & 0 & 0 & 0 & 0 & 0 & 0 & 0 & 0 \\ 0 & 0 & 0 & 0 & 0 & 0 & 0 & 0 & 0 & 1 & 0 & 0 & 1 & 0 & 0 & 1 & 0 & 0 & 0 & 0 & 0 & 0 & 0 & 0 & 0 & 0 & 0 \\ 0 & 0 & 0 & 0 & 0 & 0 & 0 & 0 & 0 & 0 & 1 & 0 & 0 & 1 & 0 & 0 & 1 & 0 & 0 & 0 & 0 & 0 & 0 & 0 & 0 & 0 & 0 \\ 0 & 0 & 0 & 0 & 0 & 0 & 0 & 0 & 0 & 0 & 0 & 1 & 0 & 0 & 1 & 0 & 0 & 1 & 0 & 0 & 0 & 0 & 0 & 0 & 0 & 0 & 0 \\ 0 & 0 & 0 & 0 & 0 & 0 & 0 & 0 & 0 & 0 & 0 & 0 & 0 & 0 & 0 & 0 & 0 & 0 & 1 & 0 & 0 & 1 & 0 & 0 & 1 & 0 & 0 \\ 0 & 0 & 0 & 0 & 0 & 0 & 0 & 0 & 0 & 0 & 0 & 0 & 0 & 0 & 0 & 0 & 0 & 0 & 0 & 1 & 0 & 0 & 1 & 0 & 0 & 1 & 0 \\ 0 & 0 & 0 & 0 & 0 & 0 & 0 & 0 & 0 & 0 & 0 & 0 & 0 & 0 & 0 & 0 & 0 & 0 & 0 & 0 & 1 & 0 & 0 & 1 & 0 & 0 & 1 \end{pmatrix}^T$$

It is easy to verify that, under the assumption, the output $O$ will be equal to $R_C = (T_{C,1,1}, T_{C,1,2}, \ldots, T_{C,3,2}, T_{C,3,3})$, where $T_C = T_A \cdot T_B$. So FloydNet can precisely express the composition of rotation transformations. For geometry tasks, it is not necessary to explicitly set the values of linear transformation matrices $M_A, M_B, M_C$, they can be implicitly learned on training.

### B.2 LIMITATIONS AND ENGINEERING OPTIMIZATION

The primary trade-off of the FloydNet architecture is its computational complexity. The global, all-pairs refinement via Pivotal Attention scales as $\mathcal{O}(N^3)$, which is inherent to the 3-WL expressive power it achieves. While this prohibits direct application to massive graphs (e.g., millions of nodes), it is well-suited for high-value reasoning tasks on dense graphs of moderate size (e.g., algorithmic reasoning, molecular modeling).

To mitigate this, we implemented a highly optimized a kernel(w.r.t. NVIDIA CUDA) optimized PyTorch version at `http://anonymous_for_submission`, which reduces memory usage from $\mathcal{O}(N^3)$ to $\mathcal{O}(N^2)$ and improves wall-clock speed by over $20\times$. This allows efficient training on graphs up to $N = 256$ on standard GPUs. Extending FloydNet to larger, sparse graphs via techniques such as sparse attention is a promising direction for future work. Figure 5 provides a detailed comparison of the runtime and GPU memory consumption between the kernel-optimized and the original implementations as the sequence length increases.

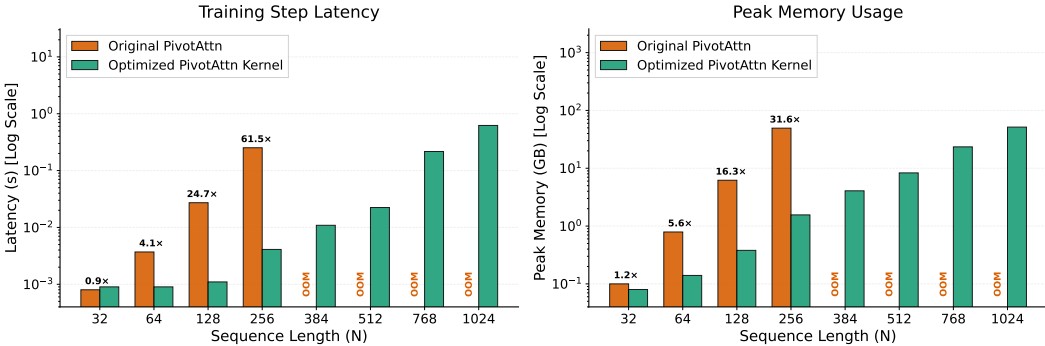

Figure 5: **Performance Comparison of Original vs. Optimized PivotAttn Kernel.** Training step latency and peak GPU memory usage are shown as sequence length $N$ increases. The original implementation quickly leads to out-of-memory (OOM) errors. At $N = 256$, the Original version requires 61.5x the runtime and 31.6x the peak memory of the Optimized version.

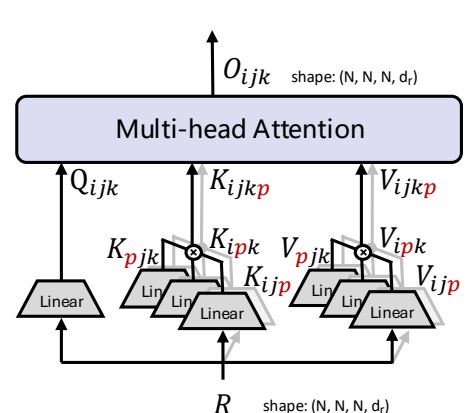

```python
def PivotAttnCore(q_ijk, k_pjk, k_ipk, k_ijp,
                  v_pjk, v_ipk, v_ijp):
    # All input tensors of shape [b,n,n,n,h,d_h]
    # come from different linear transform of R

    # Combine operation is multiplicative
    k_ijkp = torch.einsum(
             'bpjkhd,bipkhd,bijphd->bijkphd',
             k_pjk, k_ipk, k_ijp)
    v_ijkp = torch.einsum(
             'bpjkhd,bipkhd,bijphd->bijkphd',
             v_pjk, v_ipk, v_ijp)
    q_ijkp = q_ijk[:, :, :, :, None, :, :]

    a_ijkp = torch.sum(q_ijkp * k_ijkp, dim=-1)
    d = q_ijk.shape[-1] # per head embed dim
    a_ijkp = a_ijkp / (d ** 0.5) # [b,n,n,n,n,h]
    w_ijkp = torch.softmax(a_ijkp, dim=-2) # dim p

    o_ijkp = v_ijkp * w_ijkp[..., None]
    o_ijk = torch.sum(o_ijkp, dim=-3) # dim p
    return o_ijk # [b,n,n,n,h,d_h], same as input
```

Figure 6: **FloydBlock and Pivotal Attention Mechanism.(3-FloydNet)** Each `FloydBlock` updates the representation for a given pair $(i, j, k)$ by using its state as a query to attend to all possible two-hop paths that pass through an intermediate "pivot" node $p$.

### B.3 DETAILS OF THE HIGHER-ORDER K-FLOYDNET FRAMEWORK

The FloydNet architecture can be naturally generalized to a higher-order framework, k-FloydNet, which operates on k-tuples of vertices instead of pairs. This generalization allows for a systematic increase in expressive power, directly corresponding to higher levels of the Weisfeiler-Lehman hierarchy.

**Initialization of Higher-Order Relationship Tensors.** A k-FloydNet model requires an initial relationship tensor $R^{(0)} \in \mathbb{R}^{N^{\times k} \times d_r}$. For simple cases like $k = 1$ or $k = 2$, this tensor can be directly constructed from standard node features ($X$) or edge features ($E$). For higher orders ($k \geq 3$), where explicit k-tuple features are usually unavailable, we construct the initial tensor by aggregating all available lower-order relationships within it. Given a set of m-order features $r \in \mathbb{R}^{N^{\times m} \times d_m}$ (where $m \leq k$), the k-order representation is formed by applying an MLP to the concatenation of all m-tuple features that can be formed by selecting vertices from the k-tuple. The general formula is:

$$R_{v_1,v_2,...,v_k} = \mathsf{MLP}\left( \underset{1 \leq a_1 < a_2 < \cdots < a_m \leq k}{\mathsf{concat}} \left( r_{v_{a_1}, v_{a_2}, ..., v_{a_m}} \right) \right) \tag{9}$$

For example, to initialize the tensor for 3-FloydNet using graph feature $G$, node feature $X$, edge feature $E$ and 3-dimensional hyper-edge feature $H$ (which may be unavailable), we can concatenate all $\binom{3}{0} + \binom{3}{1} + \binom{3}{2} + \binom{3}{3} = 8$ possible lower-order relationships within the triplet $(i, j, k)$:

$$R_{ijk} = \mathsf{MLP}(\mathsf{concat}(G, X_i, X_j, X_k, E_{ij}, E_{ik}, E_{jk}, H_{ijk})) \tag{10}$$

**Pivotal Attention.** The Pivotal Attention mechanism, detailed for the standard FloydNet (i.e., 2-FloydNet) in Figure 1, generalizes naturally to the higher-order k-FloydNet framework as illustrated in Figure 6. The implementation is conceptually consistent, with two primary modifications:

1. **Tensor Dimensionality:** The **Query**, **Key**, and **Value** tensors are promoted from operating on pairs to operating on k-tuples. For instance, in a 3-FloydNet model, these tensors would have a shape of $(b, n, n, n, h, d_h)$, where $b$ is the batch size and $n$ is the number of nodes.

2. **Aggregation Logic:** The attention mechanism is performed at the k-th order. To update the representation for a k-tuple $(v_1, \ldots, v_k)$, the mechanism aggregates information from all pivots $w \in V$. The "path segments" for the aggregation are the $k$ different k-tuples formed by substituting each $v_i$ with $w$ in turn (e.g., $(w, v_2, \ldots, v_k), (v_1, w, \ldots, v_k)$, etc.).

**Implementations.**  In consideration of its immediate practical value, our open-source contribution primarily focuses on the implementation of the standard FloydNet. For the BREC benchmark, we developed a straightforward, proof-of-concept implementation of k-FloydNet to validate its theoretical expressiveness. Developing highly optimized kernels for the higher-order variants is a non-trivial engineering task that we leave as a direction for future work.

### B.4  COMPLEXITY OF FLOYDNET

The computational complexity of a single `FloydBlock` layer is dominated by the Pivotal Attention mechanism. Let $N$ be the number of nodes, $d_r$ be the hidden dimension, and $h$ be the number of attention heads. The head dimension is $d_h = d_r/h$.

The computation of queries, keys, and values tensors involves linear projections on the $N \times N$ relationship tensor, resulting in a complexity of $\mathcal{O}(N^2 \cdot d_r^2)$.

The Pivotal Attention, as shown in the Python code in Figure 1, involves additions and multiplications scale as $\mathcal{O}(h \cdot N^3 \cdot d_{head}) = \mathcal{O}(N^3 \cdot d_r)$. The softmax is also computed over $N$ pivots for each of the $N^2$ pairs and each of the $h$ heads, costing $\mathcal{O}(N^3 \cdot h)$.

The FFN part has a complexity of $\mathcal{O}(N^2 \cdot d_r^2)$.

Therefore, the total computational complexity per layer is $\mathcal{O}(N^3 \cdot d_r + N^2 \cdot d_r^2)$. For typical graph sizes where $N > d_r$, the complexity is effectively $\mathcal{O}(N^3 \cdot d_r)$, which is cubic in the number of nodes, analogous to the classic Floyd-Warshall algorithm.

The general form k-FloydNet operates on the $N^k$ relationship tensor (embeddings of hyper-edges). The cost of linear projections and FFN are both $\mathcal{O}(N^k \cdot d_r^2)$. The cost of Pivotal Attention is $\mathcal{O}(N^{k+1} \cdot d_r)$. The overall complexity is $\mathcal{O}(N^{k+1} \cdot d_r + N^k \cdot d_r^2)$. For typical graph sizes where $N > d_r$ , the complexity is effectively $\mathcal{O}(N^{k+1} \cdot d_r)$.

### B.5  COMPARISON WITH TRANSFORMERS AND THE CANONICAL ORDERING PROBLEM

To further clarify the positioning of our work, Table 6 provides an illustrative comparison between the standard Transformer, FloydNet, and its higher-order generalization, k-FloydNet. This comparison highlights a key conceptual point: FloydNet provides a new perspective on the fundamental gap between sequence modeling and graph reasoning.

Sequence models like the Transformer excel when given a clear, fixed order. Theoretically, any graph problem could be converted into a sequence task if one could define a canonical ordering of its nodes, maintaining a consistent sequence representation for any given graph. However, finding such an ordering is at least as difficult as the Graph Isomorphism (GI) problem, which is not known to be solvable in polynomial time. Graph Transformers attempt to work around this by using positional encodings (PEs), but this is an approximate solution that tries to inject the lost ordering and structural information back into a model that is not inherently designed for it.

FloydNet offers a more elegant and fundamental solution. Instead of attempting the intractable task of finding a single correct node permutation, FloydNet operates on the all-pairs relationship tensor. This approach lifts the computation into a higher-dimensional space where all pairwise relationships exist simultaneously. Its core mechanism, Pivotal Attention, is therefore inherently permutation-equivariant. It does not need to know the "correct" order of nodes because it processes the complete relational structure of the graph in a principled way. In this view, FloydNet can be seen not just as a new GNN, but as a principled generalization of relational reasoning to non-sequential data, without first forcing that data into a brittle, ordered format.

### B.6  A NEW SCALING PARADIGM: FROM LARGE MODELS TO COMPUTATIONALLY INTENSIVE MODELS

Our analysis of FloydNet on the TSP benchmark reveals that the architecture exhibits strong scaling properties, similar to the scaling laws observed in large Transformer models(Kaplan et al., 2020). As shown in our experiments (Figures 4d and 4e), performance on out-of-distribution instances consistently improves with both more training data and larger model capacity (e.g., more layers).

Table 6: A comparison of key architectural and theoretical properties.

| Property | Transformer | FloydNet | k-FloydNet |
|---|---|---|---|
| Core Mechanism | Self-Attention | Pivotal Attention | Higher-order Pivotal Attention |
| Operating Space | 1D Set of Node Features ($\mathcal{O}(N \cdot d)$) | 2D Tensor of Pairwise Relationships ($\mathcal{O}(N^2 \cdot d)$) | k-dim Tensor of k-tuples ($\mathcal{O}(N^k \cdot d)$) |
| Computational Complexity | $\mathcal{O}(N^2 \cdot d)$ | $\mathcal{O}(N^3 \cdot d)$ | $\mathcal{O}(N^{k+1} \cdot d)$ |
| Permutation Equivariant | No | Yes | Yes |
| Requires Positional Encoding | Yes (Essential) | No | No |
| Theoretical Expressive Power | $\preceq$ 3-WL (with strong PEs) | $\equiv$ 2-FWL | $\equiv$ k-FWL |

However, this scaling behavior in FloydNet points to a different paradigm. While Large Language Models (LLMs) scale primarily by dramatically increasing their parameter count into the billions, FloydNet achieves these performance gains with a comparatively modest number of parameters. We attribute this to its high computational intensity. Due to its $\mathcal{O}(N^3)$ complexity, the total computation performed by FloydNet for a single problem instance grows polynomially with the problem size $N$.

We hypothesize that this intrinsic, "heavy" computation is a key driver of its scaling performance. In this view, the model's effective capacity is a function of not only its parameters but also its intense, structured computation per instance. This aligns with the intuition that solving complex, System 2-style reasoning problems benefits from more dedicated computational effort, not just a larger knowledge base. While our empirical results strongly support this hypothesis, a formal analysis of the principles behind this "compute-driven" scaling is a rich area for future work. We plan to validate and expand on these findings in more complex reasoning scenarios.

## C   FORMAL DISCUSSION ON FLOYDNET'S EXPRESSIVENESS

Here we provide more formal arguments for the theoretical claims made in the main paper. Our analysis primarily relies on mapping the computation of GNNs to equivalent color refinement algorithms, a standard technique in the expressivity literature (Xu et al., 2019; Morris et al., 2019).

**Definition 1** (Color Refinement Algorithm). *A color refinement algorithm iteratively updates the 'color' (a feature vector) of entities (nodes, edges, tuple of nodes, etc.) based on the colors of related entities in their neighborhood. Two graphs are distinguishable by an algorithm if their final multisets of colors differ. A neural architecture is said to be at most as powerful as a color refinement algorithm if any two graphs distinguishable by the architecture are also distinguishable by the algorithm.*

**Theorem 3** (k-FloydNet is k-FWL). *The k-FloydNet architecture exactly matches the definition k-Folklore Weisfeiler-Lehman (k-FWL) algorithm, thus it can distinguish non-isomorphic graphs that can be distinguished by the k-FWL test, and vice versa.*

*Proof.* The expressive power of a GNN can be characterized by its analogous color refinement scheme. We consider the k-FWL test which is known to have the equivalent power as (k+1)-WL test. The k-FWL test algorithm refines the color of super-edges $\vec{e} = (v_1, v_2, \ldots, v_k)$ based on the multiset of k-tuples of colors of neighbor super-edges $\vec{e'} = (\vec{e}[i] \leftarrow p) = (v_1, v_2, \ldots, v_{i-1}, p, v_{i+1}, \ldots, v_{k-1}, v_k)$, which differs to $\vec{e}$ by replacing the $i$-th node to node $p$. We use the notation $[\![\cdot, \cdot]\!]$ to denote a tuple, and the notation $\{\!\{\cdot\}\!\}$ to denote a multiset, the formula of the color refinement iteration can be written as:

$$\mathbf{C}'_{\vec{e}} = \text{HASH}\left(\mathbf{C}_{\vec{e}}, \{\!\{[\![\mathbf{C}_{\vec{e}[1] \leftarrow p}, \mathbf{C}_{\vec{e}[2] \leftarrow p}, \ldots, \mathbf{C}_{\vec{e}[k] \leftarrow p}]\!] : p \in V\}\!\}\right) \quad (11)$$

In common implementations of k-FWL test algorithm, the color is represented by a bit vector of length $d_c$, thus $\mathbf{C} \in \mathbb{Z}_2^{N \times k \times d_c}$, where $N$ is the number of nodes.

The HASH function must have the following properties to ensure the k-FWL test's effectiveness:

1. The output hash value will change if the elements of a tuple are permuted.

2. The output hash value will not change if the elements of a multiset are permuted.

3. The output hash value will change non-linearly with respect to the any of the input values.

We will demonstrate that the k-FloydNet serves as an alternative implementation of the k-FWL test algorithm, with color representations encoded through neural network embeddings.

k-FloydNet operates on relationship tensor $\boldsymbol{R} \in \mathbb{R}^{N \times k \times d_r}$, which can be view as colors of $N^k$ super-edges, and the color is embedded in the continuous space $\mathbb{R}^{d_r}$. The Pivotal Attention module will project $\boldsymbol{R}_{\vec{e}}$ to **Query** tensor $Q_{\vec{e}}$, and project $\boldsymbol{R}_{\vec{e},p,i} = \boldsymbol{R}_{\vec{e}[i] \leftarrow p}$ to **Key** tensor $K_{\vec{e},p,i}$ and **Value** tensor $V_{\vec{e},p,i}$ for each pivot $p \in V$ and index $i \in [1, k]$. Different linear projection modules are applied for each index $i$ (as shown in Figure 1). Then a MultiheadAttention is applied to these **Query**, **Key**, and **Value** tensors to get the final output tensor $\boldsymbol{R}'_{\vec{e}}$, which we view as the refined color.

The Pivotal Attention module works as a HASH function that operates on continuous space, preserving all essential properties required by the k-FWL test:

1. The output embedding $\boldsymbol{R}'_{\vec{e}}$ will change if the indices $i \in [1, k]$ are permuted. Although the combine operator $\mathcal{C}$ is commutative, the **Key** and **Value** vectors to be combined are projected by different linear modules, ensuring that the resultant embedding reflects the distinct order.

2. The output embedding $\boldsymbol{R}'_{\vec{e}}$ will not change if the pivots $p \in V$ are permuted, due to the symmetric aggregation of MultiheadAttention.

3. The output embedding $\boldsymbol{R}'_{\vec{e}}$ will change non-linearly with respect to the any of the input embeddings $\boldsymbol{R}_{\vec{e'}}$, due to the MLPs, which can approximate any continuous function to an arbitrary degree of accuracy, according to the Universal Approximation Theorem.

So Pivotal Attention with color represented by $\boldsymbol{R}_{\vec{e}} \in \mathbb{R}^{d_r}$ is an valid implementation of the color refinement iteration of the k-FWL test algorithm. The k-FloydNet consists of multiple layers of Pivotal Attention, reflecting the multiple iterations of the color refinement process. Thus k-FloydNet is a k-FWL test algorithm.

$\square$

While having the graph distinguishing power of k-FWL, the Pivotal Attention function is fully differentiable, allowing for seamless integration into deep learning models, making it useful for enhancing the expressive capacity of GNNs.

**Theorem 4** (**Principled Long-Range Information Propagation**). *After $L$ layers, $\boldsymbol{R}_{ik}^{(L)}$ integrates information from paths of length up to $2^L$ between nodes $i$ and $k$.*

*Proof.* We proceed by induction on $L$. **Base Case ($L = 1$):** By definition, $\boldsymbol{R}_{ik}^{(1)}$ aggregates over all pivots $j$ via terms $(\boldsymbol{R}_{ij}^{(0)}, \boldsymbol{R}_{jk}^{(0)})$. Each such interaction encodes paths of length 2 of the form $i \rightsquigarrow j \rightsquigarrow k$. Thus the claim holds for $L = 1$. **Inductive Step:** Suppose that after $l - 1$ layers, $\boldsymbol{R}_{uv}^{(l-1)}$ contains information about all paths of length up to $2^{l-1}$ between $u$ and $v$. Now consider $\boldsymbol{R}_{ik}^{(l)}$. Its update depends on $\{\boldsymbol{R}_{ij}^{(l-1)}, \boldsymbol{R}_{jk}^{(l-1)}\}_{j \in V}$ through $g(\cdot, \cdot)$ and $\bigoplus$. By the induction hypothesis:

- $\boldsymbol{R}_{ij}^{(l-1)}$ aggregates paths of length up to $2^{l-1}$ from $i$ to $j$.

- $\boldsymbol{R}_{jk}^{(l-1)}$ aggregates paths of length up to $2^{l-1}$ from $j$ to $k$.

combining these two representations concatenates paths through $j$, producing paths of maximum length $2^{l-1} + 2^{l-1} = 2^l$ from $i$ to $k$. Therefore, $\boldsymbol{R}_{ik}^{(l)}$ contains information about all paths up to length $2^l$. **Conclusion:** By induction, for all $L \geq 1$, $\boldsymbol{R}_{ik}^{(L)}$ integrates information from paths of length

up to $2^L$. This exponential growth in effective receptive field contrasts sharply with MPNNs, where the receptive field expands only linearly in $L$. □

## D ADDITIONAL EXPERIMENTAL RESULTS AND DETAILS

### D.1 DETAILED RESULTS ON THE BREC BENCHMARK

To thoroughly evaluate the expressive power of our framework, we provide a detailed analysis on the BREC benchmark (Wang & Zhang, 2024). The results, presented in Table 7, compare our models against theoretical tests and other representative GNN architectures. The performance of our base model, FloydNet, aligns precisely with the theoretical 3-WL test. It correctly distinguishes all 60 Basic and 100 Extension pairs, while matching the 3-WL test's exact scores on Regular (50/140) and CFI (60/100) graphs. This provides strong evidence that FloydNet is an empirical realization of a 3-WL equivalent mechanism.

This alignment also means FloydNet inherits the limitations of the 3-WL test, particularly on certain strongly regular graphs where it fails. This creates an opportunity for synergistic model design. Subgraph-based models like KP-GNN are known to be powerful on these regular graphs, while being weaker on CFI graphs where FloydNet excels. This complementary nature suggests that combining these approaches could yield a more robust model.

To verify this, we conducted a hybrid experiment. By augmenting FloydNet with pre-computed graph features from the KP-GNN framework (denoted FloydNet-KP), the model's performance on regular graphs improves significantly. As shown in Table 7, the hybrid model correctly distinguishes 106 regular pairs, which matching KP-GNN's strength in this area while retaining FloydNet's power on CFI graphs. This boosts the total accuracy from 67.5% to 81.5%, demonstrating that FloydNet's global reasoning paradigm can be effectively combined with powerful local structure features.

The scalability of the k-FloydNet framework is also evident in the results. 3-FloydNet resolves all regular graph pairs and 80 CFI pairs, aligning its power with the 4-WL test. 4-FloydNet achieves 99.8% accuracy by correctly distinguishing 99 of the 100 CFI pairs. The single failure case is pair #356. An educated guess would be that this specific pair may be indistinguishable even by the 5-WL test; however, we did not perform rigorous verification due to its significant computational cost.

Table 7: Detailed pair distinguishing results on the BREC benchmark. The performance of k-FloydNet empirically aligns with the k-FWL test. The results are compared against theoretical tests and representative GNNs.

| Type | Model | Basic (acc. on 60) | Regular (acc. on 140) | Extension (acc. on 100) | CFI (acc. on 100) | Total (acc. on 400) |
|---|---|---|---|---|---|---|
| Theoretical Tests | 1-WL Test | 0 | 0 | 0 | 0 | 0 (0.0%) |
| | 3-WL Test | **60** | 50 | **100** | 60 | 270 (67.5%) |
| MPNNs | GIN (MPNN) | 0 | 0 | 0 | 0 | 0 (0.0%) |
| Graph Transformers | Graphormer | 16 | 12 | 41 | 10 | 79 (19.8%) |
| | GDT+RWSE | 57 | 50 | 96 | 0 | 203 (50.8%) |
| | PPGT | **60** | 50 | **100** | 24 | 234 (58.5%) |
| Subgraph GNNs | I$^2$-GNN | **60** | 100 | **100** | 21 | 281 (70.2%) |
| | KP-GNN | **60** | 106 | 98 | 11 | 275 (68.8%) |
| k-WL GNNs | PPGN | 60 | 50 | 100 | 23 | 233 (58.2%) |
| | KC-SetGNN | 60 | 50 | 100 | 1 | 211 (52.8%) |
| Ours | **FloydNet** | **60** | 50 | **100** | **60** | **270 (67.5%)** |
| | **FloydNet-KP**[*] | **60** | **106** | **100** | **60** | **326 (81.5%)** |
| | **3-FloydNet** | **60** | **140** | **100** | **80** | **380 (95.0%)** |
| | **4-FloydNet** | **60** | **140** | **100** | **99** | **399 (99.8%)** |

[*]FloydNet using KP-GNN's pre-computed graph features.

## D.2 Details on the CLRS Benchmarks

**Hint vs. No Hint.** In the CLRS benchmark, *hints* refer to the intermediate steps of algorithms, which are intended to help models better capture the underlying mechanisms and thereby enhance their reasoning ability on OOD data. When hints are used, training and inference follow an auto regressive paradigm. Prior work (Mahdavi et al., 2022) observed that disabling hints can in fact improve performance for most algorithms. We similarly find that, in the majority of tasks, hints do not provide performance gains; moreover, enabling hints incurs additional memory overhead, preventing us from increasing the depth of FloydNet. Interestingly, the subset of algorithms where hints benefit FloydNet primarily fall under Greedy methods (task scheduling, activity selection) and Dynamic Programming (maximum subarray with Kadane's algorithm, LCS length, optimal BST, and matrix chain order). Although the maximum subarray task is categorized as Divide and Conquer in CLRS, the Kadane's variant is in fact a dynamic programming method. Notably, FloydNet shows relatively weaker performance on Greedy, DP, and the D&C (but effectively DP) tasks in Table 4. We hypothesize that the additional memory cost of hints forced us to adopt a shallower FloydNet with $L = 8$ layers, leading to a performance bottleneck. We are actively exploring strategies to push beyond this limitation. For tasks trained without hints, we discarded the redundant intermediate steps and standardized the number of steps to 1. We use $L = 32$ to $L = 80$ layers when training without hints. For algorithms such as Floyd–Warshall, OOD performance consistently improves as $L$ increases. We further analyze the extrapolation capabilities of FloydNet, and the influence of using hints, by adopting the experimental setting introduced by (Minder et al., 2023). Specifically, models are trained on smaller graphs ($N = 16$ nodes) and subsequently evaluated on larger instances ($N = 160$ nodes). As detailed in Table 11, FloydNet model (without hints) exhibits nearly perfect accuracy across the larger sized test cases for BFS, Dijkstra, and MST. Conversely, the performance of the hint-augmented version is comparatively inferior on these extrapolation tasks.

**Online vs. Offline.** The CLRS framework provides both a preprocessed, finite training dataset and an on-the-fly data sampling procedure. We denote training with the finite dataset as *offline training* and training with unlimited sampled data as *online training*. As discussed earlier, FloydNet exhibits strong scaling capability, but with limited training data it tends to suffer from premature convergence, which in turn harms generalization. We trained both offline and online variants using identical hyperparameters, denoting the online version as FloydNet $_{full}$ in Table 9. Across nearly all algorithms, online training yields superior OOD test performance. For instance, in sorting tasks, performance improves from nearly perfect to consistently $100\%$, while in KMP matching, accuracy rises dramatically from 27% to 99%.

Table 8: Early-stopped validation accuracy (%) on all 30 CLRS algorithms.

| Algorithm | Deep Sets | GAT | Memnet | MPNN | PGN | FloydNet | FloydNet $_{full}$ |
|---|---|---|---|---|---|---|---|
| Activity Selector | $83.50\% \pm 0.17$ | $92.40\% \pm 0.50$ | $34.59\% \pm 2.15$ | $93.89\% \pm 0.39$ | $82.26\% \pm 0.19$ | $97.20\% \pm 0.87$ | $100.00\% \pm 0.00$ |
| Articulation Points | $99.63\% \pm 0.31$ | $100.00\% \pm 0.00$ | $16.84\% \pm 1.03$ | $100.00\% \pm 0.00$ | $100.00\% \pm 0.00$ | $100.00\% \pm 0.00$ | $100.00\% \pm 0.00$ |
| Bellman-Ford | $81.12\% \pm 0.14$ | $99.28\% \pm 0.14$ | $68.75\% \pm 0.42$ | $99.48\% \pm 0.05$ | $99.35\% \pm 0.05$ | $99.47\% \pm 0.61$ | $100.00\% \pm 0.00$ |
| BFS | $100.00\% \pm 0.00$ | $100.00\% \pm 0.00$ | $70.70\% \pm 0.09$ | $100.00\% \pm 0.00$ | $100.00\% \pm 0.00$ | $100.00\% \pm 0.00$ | $100.00\% \pm 0.00$ |
| Binary Search | $93.34\% \pm 0.41$ | $95.72\% \pm 0.17$ | $20.33\% \pm 0.28$ | $94.19\% \pm 0.12$ | $94.17\% \pm 0.08$ | $94.67\% \pm 0.74$ | $99.47\% \pm 0.67$ |
| Bridges | $99.35\% \pm 0.05$ | $100.00\% \pm 0.00$ | $96.46\% \pm 1.13$ | $100.00\% \pm 0.00$ | $100.00\% \pm 0.00$ | $100.00\% \pm 0.00$ | $100.00\% \pm 0.00$ |
| Bubble Sort | $81.51\% \pm 1.02$ | $95.44\% \pm 1.01$ | $92.64\% \pm 0.14$ | $94.53\% \pm 1.84$ | $87.17\% \pm 5.46$ | $100.00\% \pm 0.00$ | $100.00\% \pm 0.00$ |
| DAG Shortest Paths | $92.25\% \pm 0.28$ | $96.81\% \pm 0.05$ | $81.90\% \pm 0.05$ | $99.93\% \pm 0.05$ | $99.80\% \pm 0.00$ | $99.80\% \pm 0.00$ | $100.00\% \pm 0.00$ |
| DFS | $62.76\% \pm 1.26$ | $99.22\% \pm 0.64$ | $47.72\% \pm 0.45$ | $100.00\% \pm 0.00$ | $100.00\% \pm 0.00$ | $100.00\% \pm 0.00$ | $100.00\% \pm 0.00$ |
| Dijkstra | $80.34\% \pm 0.42$ | $99.22\% \pm 0.40$ | $67.38\% \pm 0.70$ | $99.67\% \pm 0.14$ | $99.28\% \pm 0.05$ | $100.00\% \pm 0.00$ | $100.00\% \pm 0.00$ |
| Find Max. Subarray | $91.41\% \pm 0.22$ | $95.00\% \pm 0.32$ | $27.91\% \pm 0.08$ | $95.13\% \pm 0.37$ | $95.30\% \pm 0.16$ | $95.63\% \pm 1.42$ | $99.90\% \pm 0.00$ |
| Floyd-Warshall | $35.79\% \pm 0.04$ | $87.28\% \pm 0.09$ | $31.29\% \pm 0.04$ | $89.14\% \pm 0.03$ | $88.70\% \pm 0.15$ | $98.97\% \pm 0.06$ | $99.87\% \pm 0.06$ |
| Graham Scan | $87.66\% \pm 0.24$ | $97.85\% \pm 0.11$ | $53.53\% \pm 1.58$ | $98.45\% \pm 0.15$ | $89.06\% \pm 0.27$ | $99.20\% \pm 0.28$ | $100.00\% \pm 0.00$ |
| Heapsort | $81.84\% \pm 0.33$ | $87.24\% \pm 2.23$ | $54.04\% \pm 0.28$ | $94.27\% \pm 0.11$ | $90.36\% \pm 0.67$ | $98.10\% \pm 0.00$ | $100.00\% \pm 0.00$ |
| Insertion Sort | $89.58\% \pm 0.28$ | $95.18\% \pm 0.58$ | $94.40\% \pm 0.14$ | $96.74\% \pm 0.19$ | $84.57\% \pm 0.82$ | $99.50\% \pm 0.42$ | $100.00\% \pm 0.00$ |
| Jarvis' March | $72.82\% \pm 0.42$ | $98.38\% \pm 0.16$ | $37.92\% \pm 6.61$ | $97.94\% \pm 0.25$ | $88.34\% \pm 0.36$ | $98.10\% \pm 0.00$ | $100.00\% \pm 0.00$ |
| KMP Matcher | $98.03\% \pm 0.21$ | $99.76\% \pm 0.08$ | $9.67\% \pm 0.00$ | $99.87\% \pm 0.05$ | $94.14\% \pm 0.99$ | $100.00\% \pm 0.00$ | $100.00\% \pm 0.00$ |
| LCS Length | $69.24\% \pm 0.36$ | $77.00\% \pm 0.19$ | $67.69\% \pm 0.24$ | $77.88\% \pm 0.42$ | $69.19\% \pm 0.04$ | $100.00\% \pm 0.00$ | $100.00\% \pm 0.00$ |
| Matrix Chain Order | $94.46\% \pm 0.02$ | $99.37\% \pm 0.03$ | $93.91\% \pm 0.10$ | $99.12\% \pm 0.04$ | $99.21\% \pm 0.03$ | $99.50\% \pm 0.10$ | $99.93\% \pm 0.06$ |
| Minimum | $97.59\% \pm 0.11$ | $97.74\% \pm 0.21$ | $95.56\% \pm 0.10$ | $97.64\% \pm 0.05$ | $97.07\% \pm 0.14$ | $99.50\% \pm 0.10$ | $100.00\% \pm 0.00$ |
| MST-Kruskal | $83.79\% \pm 2.01$ | $97.93\% \pm 0.25$ | $64.65\% \pm 0.95$ | $99.71\% \pm 0.17$ | $99.12\% \pm 0.08$ | $100.00\% \pm 0.00$ | $100.00\% \pm 0.00$ |
| MST-Prim | $74.61\% \pm 0.32$ | $98.37\% \pm 0.14$ | $74.09\% \pm 0.28$ | $99.02\% \pm 0.09$ | $97.79\% \pm 0.14$ | $98.30\% \pm 2.94$ | $100.00\% \pm 0.00$ |
| Naïve String Match | $100.00\% \pm 0.00$ | $100.00\% \pm 0.00$ | $9.91\% \pm 0.20$ | $100.00\% \pm 0.00$ | $50.33\% \pm 0.08$ | $100.00\% \pm 0.00$ | $100.00\% \pm 0.00$ |
| Optimal BST | $92.02\% \pm 0.14$ | $93.30\% \pm 0.49$ | $90.86\% \pm 0.40$ | $93.88\% \pm 0.11$ | $93.20\% \pm 0.27$ | $96.30\% \pm 1.27$ | $98.53\% \pm 1.23$ |
| Quickselect | $42.30\% \pm 0.92$ | $83.82\% \pm 1.86$ | $6.56\% \pm 0.25$ | $88.74\% \pm 0.78$ | $54.02\% \pm 0.17$ | $99.60\% \pm 0.00$ | $99.97\% \pm 0.06$ |
| Quicksort | $79.69\% \pm 1.12$ | $92.97\% \pm 0.40$ | $93.16\% \pm 0.24$ | $95.70\% \pm 0.40$ | $54.30\% \pm 1.42$ | $100.00\% \pm 0.00$ | $100.00\% \pm 0.00$ |
| Segments Intersect | $77.49\% \pm 0.12$ | $90.82\% \pm 0.16$ | $71.57\% \pm 1.08$ | $93.84\% \pm 0.20$ | $78.32\% \pm 0.18$ | $96.07\% \pm 0.23$ | $99.87\% \pm 0.12$ |
| SCC | $89.52\% \pm 1.23$ | $100.00\% \pm 0.00$ | $70.57\% \pm 1.43$ | $100.00\% \pm 0.00$ | $99.93\% \pm 0.05$ | $100.00\% \pm 0.00$ | $100.00\% \pm 0.00$ |
| Task Scheduling | $99.16\% \pm 0.04$ | $99.80\% \pm 0.04$ | $84.80\% \pm 0.09$ | $100.00\% \pm 0.00$ | $99.06\% \pm 0.08$ | $100.00\% \pm 0.00$ | $100.00\% \pm 0.00$ |
| Topological Sort | $47.23\% \pm 0.81$ | $100.00\% \pm 0.00$ | $8.30\% \pm 0.50$ | $100.00\% \pm 0.00$ | $100.00\% \pm 0.00$ | $100.00\% \pm 0.00$ | $100.00\% \pm 0.00$ |
| Overall average | 80.93% | 95.66% | 57.92% | 96.63% | 89.47% | 99.00% | 99.92% |

Table 9: Test performance (%) of all models on all 30 CLRS algorithms, reported as mean ± std over 3 runs.

| Algorithm | Triplet-GMPNN | RT | G-ForgetNet | RANR | FloydNet | FloydNet $_{full}$ |
|---|---|---|---|---|---|---|
| Activity Selector | 95.18% ± 0.45 | 87.72% ± 2.7 | 99.03% ± 0.10 | 95.23% ± 0.71 | 92.27% ± 0.80 | 95.30% ± 1.56 |
| Articulation Points | 91.04% ± 0.92 | 34.15% ± 14.6 | 97.97% ± 0.46 | 26.32% ± 27.34 | 93.03% ± 1.37 | 100.00% ± 0.00 |
| Bellman-Ford | 97.39% ± 0.19 | 94.24% ± 1.5 | 99.18% ± 0.11 | 96.00% ± 0.38 | 94.10% ± 0.79 | 100.00% ± 0.00 |
| BFS | 99.93% ± 0.03 | 99.14% ± 0.7 | 99.96% ± 0.01 | 100.0% ± 0.0 | 100.00% ± 0.00 | 100.00% ± 0.00 |
| Binary Search | 77.58% ± 2.35 | 81.48% ± 6.7 | 85.96% ± 1.59 | 64.71% ± 6.79 | 75.53% ± 5.40 | 94.47% ± 5.26 |
| Bridges | 97.70% ± 0.34 | 37.88% ± 11.8 | 99.43% ± 0.15 | 72.22% ± 12.66 | 99.45% ± 0.21 | 100.00% ± 0.00 |
| Bubble Sort | 80.51% ± 9.10 | 38.22% ± 13.0 | 83.19% ± 2.59 | 95.78% ± 0.40 | 89.70% ± 17.41 | 100.00% ± 0.00 |
| DAG Shortest Paths | 98.19% ± 0.30 | 96.61% ± 1.6 | 99.37% ± 0.03 | 96.40% ± 1.47 | 97.90% ± 0.00 | 100.00% ± 0.00 |
| DFS | 100.0% ± 0.00 | 39.23% ± 10.5 | 74.31% ± 5.03 | 100.0% ± 0.0 | 100.00% ± 0.00 | 100.00% ± 0.00 |
| Dijkstra | 96.05% ± 0.60 | 91.20% ± 5.8 | 99.14% ± 0.06 | 95.04% ± 1.62 | 97.07% ± 2.22 | 97.67% ± 2.55 |
| Find Max. Subarray | 76.36% ± 0.43 | 66.52% ± 3.7 | 78.97% ± 0.70 | 83.53% ± 2.17 | 68.60% ± 2.69 | 86.20% ± 0.95 |
| Floyd-Warshall | 48.52% ± 1.04 | 31.59% ± 7.6 | 56.32% ± 0.86 | 27.49% ± 6.95 | 81.40% ± 2.21 | 96.20% ± 1.93 |
| Graham Scan | 93.62% ± 0.91 | 74.15% ± 7.4 | 97.67% ± 0.14 | 76.20% ± 4.51 | 96.20% ± 0.00 | 99.63% ± 0.15 |
| Heapsort | 49.13% ± 10.35 | 32.96% ± 14.8 | 57.47% ± 6.08 | 93.07% ± 1.03 | 99.00% ± 0.00 | 100.00% ± 0.00 |
| Insertion Sort | 87.21% ± 2.80 | 89.43% ± 9.0 | 98.40% ± 0.21 | 93.00% ± 1.77 | 95.35% ± 0.35 | 100.00% ± 0.00 |
| Jarvis' March | 91.01% ± 1.30 | 94.57% ± 2.2 | 88.53% ± 2.96 | 91.83% ± 1.77 | 99.00% ± 0.00 | 99.50% ± 0.14 |
| KMP Matcher | 19.51% ± 4.57 | 0.03% ± 0.1 | 12.45% ± 3.12 | 4.54% ± 2.60 | 27.37% ± 9.08 | 99.43% ± 0.38 |
| LCS Length | 80.51% ± 1.84 | 83.32% ± 4.1 | 85.43% ± 0.47 | 66.91% ± 2.53 | 91.70% ± 4.61 | 95.70% ± 1.60 |
| Matrix Chain Order | 91.68% ± 0.59 | 91.89% ± 1.2 | 91.08% ± 0.51 | 25.12% ± 1.86 | 89.23% ± 0.64 | 91.50% ± 1.41 |
| Minimum | 98.43% ± 0.01 | 95.28% ± 2.0 | 99.26% ± 0.08 | 96.92% ± 0.09 | 99.07% ± 0.06 | 100.00% ± 0.00 |
| MST-Kruskal | 89.93% ± 0.43 | 64.91% ± 11.8 | 91.25% ± 0.40 | 67.29% ± 0.93 | 93.47% ± 1.36 | 99.63% ± 0.21 |
| MST-Prim | 87.64% ± 1.79 | 85.77% ± 7.9 | 95.19% ± 0.33 | 86.60% ± 4.42 | 90.53% ± 12.05 | 99.67% ± 0.31 |
| Naïve String Match | 78.67% ± 4.99 | 65.01% ± 32.3 | 97.02% ± 0.77 | 93.71% ± 2.26 | 100.00% ± 0.00 | 100.00% ± 0.00 |
| Optimal BST | 73.77% ± 1.48 | 74.40% ± 2.6 | 83.58% ± 0.49 | 36.04% ± 12.55 | 78.00% ± 1.13 | 82.87% ± 0.96 |
| Quickselect | 0.47% ± 0.25 | 19.18% ± 17.3 | 6.30% ± 0.85 | 87.08% ± 2.21 | 83.70% ± 6.36 | 80.33% ± 8.05 |
| Quicksort | 85.69% ± 4.53 | 39.42% ± 13.2 | 73.28% ± 6.25 | 94.73% ± 0.63 | 99.80% ± 0.10 | 100.00% ± 0.00 |
| Segments Intersect | 97.64% ± 0.09 | 84.94% ± 2.6 | 99.06% ± 0.39 | 97.30% ± 0.29 | 95.87% ± 0.12 | 99.90% ± 0.10 |
| SCC | 43.43% ± 3.15 | 28.59% ± 15.2 | 53.53% ± 2.48 | 48.43% ± 8.01 | 89.27% ± 1.75 | 90.07% ± 0.57 |
| Task Scheduling | 87.25% ± 0.35 | 82.93% ± 1.8 | 84.55% ± 0.35 | 88.08% ± 1.30 | 87.23% ± 7.40 | 91.05% ± 1.75 |
| Topological Sort | 87.27% ± 2.67 | 80.62% ± 17.5 | 99.92% ± 0.02 | 74.00% ± 8.18 | 100.00% ± 0.00 | 100.00% ± 0.00 |
| **Overall average** | 80.04% | 66.18% | 82.89% | 75.78% | 90.13% | 96.64% |

Table 10: Comparison of FloydNet $_{full}$ performance with and without hints on the CLRS-30 test set. Algorithms are grouped by the impact of hints. The column shows the accuracy point difference (With Hint - No Hint), with positive (beneficial) changes in green and negative (detrimental) changes in red. OOM indicates an Out-of-Memory error.

| Algorithm | With Hint (%) | No Hint (%) | |
|---|---|---|---|
| **Group 1: Algorithms Where Hints Improve Performance** | | | |
| Find Max. Subarray | 86.20 | 80.57 | 5.63 |
| LCS Length | 95.70 | 86.10 | 9.60 |
| Matrix Chain Order | 91.50 | 85.15 | 6.35 |
| Optimal BST | 82.87 | 74.50 | 8.37 |
| Activity Selector | 95.30 | 93.40 | 1.90 |
| DFS | 100.00 | 75.10 | 24.90 |
| **Group 2: Algorithms Where Hints Degrade Performance or Cause OOM** | | | |
| Binary Search | 24.70 | 94.47 | −69.77 |
| Floyd-Warshall | 64.20 | 96.20 | −32.00 |
| Graham Scan | 88.80 | 99.63 | −10.83 |
| Quickselect | 52.60 | 80.33 | −27.73 |
| Segments Intersect | 79.10 | 99.90 | −20.80 |
| Bellman-Ford | 99.60 | 100.00 | −0.40 |
| DAG Shortest Paths | 96.80 | 100.00 | −3.20 |
| Dijkstra | 97.67 | 99.40 | −1.73 |
| Articulation Points | OOM | 100.00 | — |
| Bridges | OOM | 100.00 | — |
| Bubble Sort | OOM | 100.00 | — |
| Heapsort | OOM | 100.00 | — |
| Jarvis' March | OOM | 99.50 | — |
| KMP Matcher | OOM | 99.43 | — |
| MST-Kruskal | OOM | 99.63 | — |
| Quicksort | OOM | 100.00 | — |
| SCC | OOM | 90.07 | — |
| **Group 3: Algorithms with Saturated or Neutral Performance** | | | |
| BFS | 100.00 | 100.00 | 0.00 |
| Insertion Sort | 99.50 | 100.00 | −0.50 |
| Minimum | 99.90 | 100.00 | −0.10 |
| MST-Prim | 99.67 | 99.65 | 0.02 |
| Naïve String Match | 100.00 | 100.00 | 0.00 |
| Task Scheduling | 91.05 | 91.50 | −0.45 |
| Topological Sort | 100.00 | 99.75 | 0.25 |

Table 11: Node accuracy of models on Erdős-Rényi (ER) random graphs ($N = 160$). Algorithms denoted with (H) utilize hints.

| | BFS | BFS (H) | DFS | DFS (H) | Dijkstra | Dijkstra (H) | MST | MST (H) |
|---|---|---|---|---|---|---|---|---|
| **GIN(E)** | 99.3 | 95.1 | 19.7 | 20 | 84.3 | 53.3 | 77.6 | 49.5 |
| **PGN** | 99.5 | 99.6 | 29.9 | 26.9 | 97.2 | 92 | 84.6 | 75.6 |
| **RecGNN** | 99.5 | 99.3 | 18.7 | 13.5 | 76 | 25 | 66.6 | 25.7 |
| **FloydNet** | 100.0 | 100.0 | 60.6 | 26.4 | 99.3 | 75.2 | 97.6 | 38.9 |

**Model detail.** Table 12 reports the number of parameters and training time for FloydNet on the CLRS benchmark. The model is implemented with *Jax*(Bradbury et al., 2018) based on the official codebase of CLRS. The model is configured with head_dim = 64, num_heads = 6, batch_size=32, and trained for 40K iterations.

Table 12: Model parameters, memory consumption, and training time for 40K iterations on CLRS and TSP benchmark.

| | CLRS Benchmark | | TSP Benchmark | | |
|---|---|---|---|---|---|
| | **Params** | **Time** | **Params** | **GPU Mem.** | **Time** |
| **Depth** | (M) | (min) | (M) | (GB) | (min) |
| 8 | 20.2 | 33.3 | 17.8 | 8.8 | 19.7 |
| 16 | 37.9 | 46.7 | 35.5 | 12.8 | 30.8 |
| 32 | 73.3 | 86.7 | 70.9 | 23.2 | 53.0 |
| 64 | 144.2 | 160.0 | 141.7 | 46.7 | 100.0 |
| 96 | 215.0 | 233.3 | 212.6 | 67.5 | 146.7 |

**LLMs on CLRS.** McLeish et al. (2024) recently benchmarked mainstream Large Language Models (LLMs) on the CLRS algorithmic reasoning suite. Their central finding is that ChatGPT, when augmented with a code interpreter, can outperform Graph Neural Network (GNN) models specifically designed for these tasks. However, we argue that these results are not directly comparable to our approach. The LLM's success hinges on its ability to generate and execute code, effectively outsourcing the algorithmic reasoning to an external, deterministic interpreter. This methodology fundamentally differs from our goal of learning the algorithmic process end-to-end within the model's architecture itself.

### D.3 DETAILS ON THE TSP BENCHMARK

**General vs. metric TSP** A crucial distinction in the Traveling Salesman Problem (TSP) exists between the general(non-metric) TSP and its more structured special case, the metric TSP. The metric TSP, which includes the common Euclidean TSP, requires that all edge weights obey the triangle inequality (i.e., for any three nodes $i, j, k$, the direct path is shortest: $w_{ik} \leq w_{ij} + w_{jk}$). This property provides a strong geometric structure that many algorithms exploit. For example, a polynomial-time approximation scheme (PTAS) exists for Euclidean TSP (Arora, 1998), making it significantly more tractable.

The **general TSP**, in contrast, places no such restrictions on edge weights, making it a fundamentally harder and more challenging combinatorial problem. Success on the general TSP serves as a more robust test of a model's core reasoning abilities, as it cannot rely on geometric heuristics. In our work, we provide a comprehensive analysis by testing and comparing our model's performance on both variants, with a particular focus on the general TSP to demonstrate the robustness of our approach.

**Solvers for Ground Truth and Baseline**

- **Concorde:** Concorde(Aldous & Percus, 2003) is widely regarded as the most effective exact TSP solver for general instances. It uses a sophisticated branch-and-cut algorithm built upon integer

linear programming. While it can find the provably optimal solution, its runtime can be exponential in the worst case. We use Concorde to generate the ground-truth labels (the optimal tour) for our dataset.

- **Linkern (Chained Lin-Kernighan):** The Lin-Kernighan (LK) heuristic(Lin & Kernighan, 1973) is a powerful local search algorithm for TSP. Linkern is a highly effective implementation of an enhanced version called Chained Lin-Kernighan, which helps it escape local minima. It is one of the best-performing heuristic methods, capable of finding near-optimal solutions very quickly. We use Linkern as a strong, non-learned baseline to compare against FloydNet's performance.

**Data Generation and Filtering Pipeline** Our data generation process is designed to create challenging and unambiguous test cases. First, we generate instances with the number of nodes $N$ ranging from 10 to 200. We create two types of instances: **Euclidean TSP**, for which we generate $N$ unique 2D integer coordinates $(X, Y)$ uniformly from $[1, 100]$, and **general TSP** (also known as explicit TSP), for which we directly generate an $N \times N$ distance matrix with integer weights uniformly sampled from $[1, 100]$. For Euclidean instances, the raw coordinates are directly passed to TSP solvers, so they can leverage the geometric structure of the problem to optimize the solution. But the coordinates are converted into a distance matrix before feeding to our model, to ensure it does not access the geometric information.

Second, we solve each instance to find its ground-truth optimal tour using the state-of-the-art Concorde solver. To handle difficult cases, we implemented a robust retry mechanism that systematically alternates solver seeds and search strategies (e.g., BFS, DFS). This computationally intensive stage utilized over 1 million CPU-core-hours of computation, yielding a dataset of 40 million solved instances.

Third, to ensure each problem has a single, unambiguous optimal solution, we perform a rigorous uniqueness-filtering step. An instance is retained only if the optimal tour found by Concorde is unique, which we verify by slightly perturbing the problem and re-solve it to see if the output path remains the same. This filtering ensures our evaluation metrics are well-defined. We found that TSP problems are very likely to have multiple optimal solutions, especially for Euclidean instances and large graphs (due to combinatorial explosion). About 90% of the instances are filtered out due to having ambiguous optimal solutions. After that, we run Linkern solver on each instance to produce high-quality approximate solutions, which provide a baseline for comparison with our model's performance. This step also takes over 1 million CPU-core-hours, yielding a dataset of 4 million instances. Figure 7 illustrates the distribution of the filtered dataset.

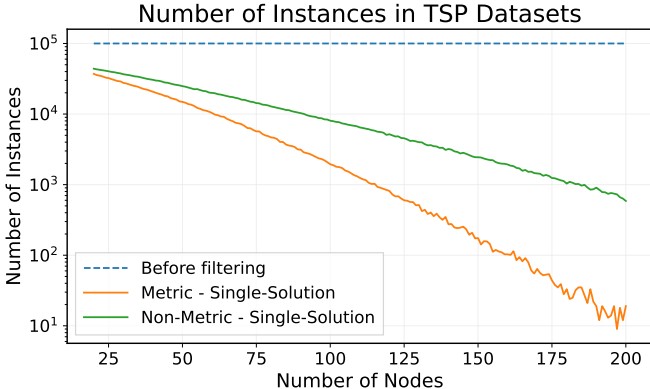

Figure 7: The data distribution of Test Datasets

Finally, we partition the dataset into a training/validation set with $N \leq 100$ and a test set with $100 < N \leq 200$. This split is designed to evaluate the model's ability to generalize to larger problem sizes than seen during training.

**Baseline and Metrics.** Dwivedi et al. (2023) established TSP datasets and baseline. However, this dataset only contains metric TSP instances, which we consider insufficient to thoroughly evaluate capabilities and limits of our model. Moreover, existing results on these datasets predominantly use

F1 score as the evaluation metric. As visualized in Figure 8, even when a model completely fails to produce an optimal TSP tour, its F1 score can still exceed 90%, indicating that F1 score lacks sufficient discriminative power. To address this, we adopt the *optimality rate* as the evaluation metric for model predictions, defined as whether the predicted tour is a valid Hamiltonian cycle with a total length equal to the TSP solution produced by the Concorde solver. We also report the detail gap on Table 13

**Model Details**   Despite the relatively high spatiotemporal complexity of FloydNet, a well-optimized kernel makes end-to-end training feasible. Table 12 reports the number of parameters, memory consumption, and training time for FloydNet on the TSP dataset. The model is configured with head_dim = 64, num_heads = 6, batch_size=1, and bfloat16 precision, and is trained for 40K iterations with a gradient accumulation factor of 8.

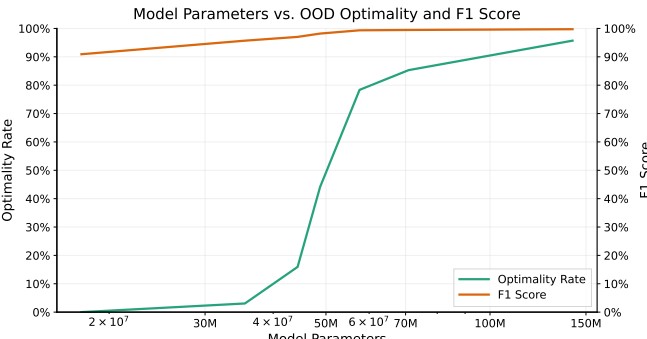

Figure 8: As the model size increases, the optimality rate rises from 0% to 96%, while the F1 score remains above 90%, indicating that F1 score is not a reliable metric for TSP tasks.

Table 13: Detailed Performance on TSP-100 and TSP-200 benchmarks. We report the gap to the Linkern solver, the percentage of valid tours found (Valid Rate), the optimality gap to the ground truth (Gap), and the average inference time. For multi-solution settings, we evaluate performance with different sampling budgets ($k = 1, 5, 10$).

| Setting | Type | Samples | TSP-100 | | | | TSP-200 | | | |
|---|---|---|---|---|---|---|---|---|---|---|
| | | | Linkern Gap (%) | Valid (%) | Gap (%) | Time (s) | Linkern Gap (%) | Valid (%) | Gap (%) | Time (s) |
| **Single Solution** | Metric | 1 | 0.001 | 98.11 | 0.001 | 0.07 | 0.017 | 89.23 | 0.002 | 3.03 |
| | Non-Metric | 1 | 0.033 | 97.03 | 0.001 | - | 0.375 | 92.63 | 0.000 | - |
| **Multi Solution** | Metric | 1 | 0.002 | 99.99 | 0.025 | 3.13 | 0.019 | 98.94 | 0.075 | 14.83 |
| | | 5 | - | 99.99 | 0.002 | - | - | 99.06 | 0.011 | - |
| | | 10 | - | 99.99 | 0.001 | - | - | 99.01 | 0.004 | - |
| | Non-Metric | 1 | 0.029 | 99.83 | 0.062 | - | 0.290 | 99.67 | 0.060 | - |
| | | 5 | - | 99.85 | 0.004 | - | - | 99.69 | 0.003 | - |
| | | 10 | - | 99.85 | 0.001 | - | - | 99.71 | 0.001 | - |

### D.4   RESULTS ON REAL-WORLD DATASETS

**ZINC.**   FloydNet achieves a new state-of-the-art Mean Absolute Error (MAE) on both the ZINC-subset and full datasets (Table 14). This showcases its effectiveness at capturing the global molecular structures that determine chemical properties, outperforming models that rely on more local aggregation.

**LRGB.**   On the Long-Range Graph Benchmark (LRGB), FloydNet excels at tasks requiring precise pairwise reasoning, setting a new state-of-the-art on link prediction (PCQM-Contact) and node classification (COCO-SP), as shown in Table 5. While its performance on the remaining graph-level tasks is competitive, our analysis of the learning curves (Figure 9) reveals a degree of overfitting. This indicates potential for further improvement through extended regularization tuning (e.g., adjusting batch sizes or weight decay), a direction we leave for future work due to computational constraints.

Table 14: MAE on the ZINC dataset. Lower is better.

| Class | Model | ZINC | |
| | | Subset | Full |
| --- | --- | --- | --- |
| GNNs | MPNN | .138 | .030 |
| | Subgraph GNN | .110 | .028 |
| | Local 2-GNN | .069 | .024 |
| GTs | Graphormer (Zhang et al., 2023) | .081 | .025 |
| | ET$_{+RRWP}$(Müller et al., 2024) | .059 | .024 |
| Ours | **FloydNet** | **.058** | **.016** |

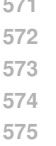
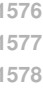
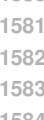
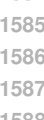
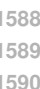
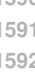

Figure 9: Training and validation loss on LRGB tasks. The curves demonstrate overfitting: as training epochs increase, training loss consistently declines while validation loss begins to rise. A full search for regularization hyperparameters to mitigate this was limited by computational resources.

# E  THE USE OF LARGE LANGUAGE MODELS (LLMs)

During the preparation of this manuscript, the authors utilized Large Language Models (LLMs) to assist in editing and refining the language and formatting to improve clarity and readability. All scientific contributions, including the core ideas, theoretical arguments, experimental design, and analysis of results, were conceived and executed independently by the authors.

Additionally, in the literature review phase, the authors referenced summaries generated by AI-powered research tools (such as Deep Research). The final selection, critical analysis, and synthesis of all cited literature presented in this paper were performed entirely by the authors.

Indeed, this usage is consistent with the paper's motivation: the authors maintain that current LLMs, while excellent assistants, do not yet possess the independent, System 2-style reasoning required to produce this research on their own.

