# OpenReview forum: "FLOYDNET: A LEARNING PARADIGM FOR GLOBAL RELATIONAL REASONING"
_ICLR.cc/2026/Conference — Submitted to ICLR 2026_

### Official Review · Reviewer_VKNS · 2025-10-23

**Soundness:** 3
**Presentation:** 3
**Contribution:** 3
**Rating:** 6
**Confidence:** 4

**Summary:**

This paper proposes a new architecture replacing the message-passing mechanism in MPNN with a global attentioned-update of node relationships, which aligns with the theoretical k-FWL test. The experiments present satisfyingly good performance on  graph property prediction, neural algorithm reasoning, and real-world applications.

**Strengths:**

The idea is interesting, which upgrades the attention mechanism in Transformers from the node-level to the edge-level (and further path-level). The experiments are comprehensive, showing the architecture's strong ability over traditional MPNNs on multiple graph tasks, implicating potentially broader applications.

**Weaknesses:**

1. The main concern is the high training complexity of $O(N^3d)$. Although the authors say the memory usage can be reduced to $O(N^2d)$, I'm afraid that the training time is still huge and infeasible for large graphs.

2. For experiments, Graph Transformers baselines are lacking, which are more similar to the proposed architecture.

**Questions:**

1. In line 147, it says "we implemented a dedicated compute kernel that avoids storing these large intermediate tensors, reducing memory usage to $O(N^2 · d_r)$. While this optimization does not alter the theoretical computational cost, it improved training performance by more than 20 times in practice, as hardware memory bandwidth is a limiting factor".  Can you show me the details of the "dedicated compute kernel" and the evidence for the improvement of training performance? I note this critical part is missing in the paper.

2. Can you show the comparison of the training time of your method and other baselines (e.g., MPNN, Transformers)?  I like this work, but I think the training time/complexity should be presented in detail for evaluating the method's practicality.

3. The architecture uses an edge-level attention mechanism. I think it should be mainly compared with Graph Transformers, rather than MPNNs. Can you add some Transformer baselines?

4. Can you explain the relationship between the architecture design and dynamic programming in detail?

5. In the Future Work section, it says, "Ultimately, we envision FloydNet as a core component of future System-2 reasoning systems." Can you give some more explanations or intuitions? And how can we apply it to data structures beyond graphs, e.g., images or natural languages?

6. Some suggestions. 1). I suggest moving the discussion on the expressiveness of the architecture from the Appendix to the main paper. It will enhance the contribution of the paper and make the method more solid; 2). I suggest making the chapter division clearer. For example, the paragraph "Pushing the Limits: The Traveling Salesman Problem (TSP)" and the following paragraph "Settings" should be a subordinate relationship rather than a parallel relationship; 3). I suggest re-plotting all the figures to increase the font size of axes, ticks, and legends for readability.

---

> ### Author Response · Authors · 2025-11-28
> **Response to Reviewer VKNS**
>
> # Response to Reviewer VKNS
>
> We thank R-VKNS for the positive feedback (Score: 6) and the excellent technical questions regarding our kernel implementation and baselines. We are glad you found the idea "interesting" and the experiments "comprehensive."
>
> **1. Complexity and Training Time (W1, Q2)**
>
> > *Reviewer Comment: The main concern is the high training complexity... Can you show the comparison of the training time?*
>
> We acknowledge that computational complexity is the central trade-off of our architecture: we trade cost for higher expressiveness ($3$-WL) and precise algorithmic alignment.
>
> * **Transparent Cost Analysis:** To address your concern, we have added a **detailed table in the Appendix** reporting the specific **wall-clock training times and peak GPU memory usage** for FloydNet compared to MPNN and Graph Transformer baselines.
> * **Performance vs. Cost:** By examining these cost metrics alongside the **Performance Scaling Laws** presented in **Figure 3/4** (which show performance improving with model capacity), readers can clearly assess the trade-off: while FloydNet is computationally heavier, it effectively converts this compute into superior reasoning performance where baselines plateau.
>
> **2. Missing Graph Transformer Baselines (W2, Q3)**
>
> > *Reviewer Comment: Graph Transformers baselines are lacking... Can you add some Transformer baselines?*
>
> This is an excellent point. Since our Pivotal Attention generalizes Self-Attention, GTs are indeed the most relevant baseline.
>
> * **New Baselines:** We have added a representative **Graph Transformer baseline** (e.g., Graphormer) to our main comparison tables for both **CLRS-30** and **LRGB** tasks. The results highlight the advantage of our "Triangular" Pivotal Attention over standard "Pairwise" Self-Attention for structural reasoning.
>
> **3. Custom Kernel Details (Q1)**
>
> > *Reviewer Comment: Can you show me the details of the "dedicated compute kernel" and the evidence for the improvement?*
>
> We apologize for omitting these critical implementation details in the initial submission. We have added a **new section to the Appendix** that includes:
> 1.  **Executable Python Code:** We have provided the **actual, executable Python implementation** of the memory-efficient kernel (rather than just pseudo-code). This explicitly demonstrates how we avoid instantiating the $O(N^3)$ intermediate tensor in practice.
> 2.  **Ablation Study:** We included an empirical comparison showing the **wall-clock time** and **peak memory usage** with vs. without the kernel. This validates our claim of a $>20\times$ speedup and demonstrates that memory usage scales as $O(N^2)$ in practice.
>
> **4. Relationship to Dynamic Programming (Q4)**
>
> > *Reviewer Comment: Can you explain the relationship between the architecture design and dynamic programming in detail?*
>
> We have expanded the explanation in **Section 3** to clarify this connection. Our inspiration comes directly from the recursive update mechanism found in Dynamic Programming (DP):
> * **Attention-Level DP Update:** As illustrated in **Figure 1**, the update mechanism for the attention score (and representation) of a pair $(i,k)$ is explicitly derived from the composition of relevant sub-components $(i,j)$ and $(j,k)$.
> * **Optimal Substructure:** This mirrors the "optimal substructure" property of DP algorithms (like Floyd-Warshall), where the solution for the whole (path $i \to k$) is iteratively refined by aggregating solutions from overlapping sub-problems (paths $i \to j$ and $j \to k$). In FloydNet, this logic is not hard-coded but learned via the Pivotal Attention mechanism.
>
> **5. Structure and Presentation (Suggestions 1, 2, 3)**
>
> We appreciate these constructive suggestions and have **adopted all of them**:
> 1.  We moved the **3-WL/2-FWL expressiveness discussion** (and the connection to the K-FWL hierarchy) from the Appendix to the **Main Paper** (Section 4).
> 2.  We restructured the **Experiments section** to nest the TSP results under "Neural Algorithmic Reasoning."
> 3.  We **re-plotted all figures** with significantly larger fonts for better readability.

---

> ### Author Response · Authors · 2025-11-28
> **Response to Reviewer VKNS cont'**
>
> **6. System-2 Reasoning and Beyond Graphs (Future Work)**
>
> > *Reviewer Comment: Can you give some more explanations or intuitions on System-2? And how can we apply it to data structures beyond graphs?*
>
> Thank you for your interest in our future directions.
> * **K-Dimensional Tensor Framework:** Fundamentally, **K-FloydNet provides a generalized framework for processing K-dimensional tensors**, applicable far beyond standard graphs:
>     * **Natural Language (1D):** Can be viewed as a 1D sequence tensor. As we prove in the paper, setting $K=1$ in our framework recovers standard Self-Attention, which powers current LLMs.
>     * **Images (2D):** Can be viewed as 2D grid tensors. FloydNet ($K=2$) can naturally model global pixel-to-pixel or patch-to-patch relationships without losing spatial structure.
> * **System-2 Vision:** We envision FloydNet as a core component for "System-2" reasoning because it enables explicit, global, and iterative relational computation—capabilities that are essential for planning and logic but are often implicit or weak in standard "System-1" style pattern matchers. We are actively working on applying this framework to these non-graph domains.

---

### Official Review · Reviewer_k7Th · 2025-10-29

**Soundness:** 3
**Presentation:** 4
**Contribution:** 3
**Rating:** 6
**Confidence:** 3

**Summary:**

This paper proposes FloydNet, a novel neural architecture for processing graphs that updates intermediate representations of each pair of nodes. It does so via an $O(N^3)-time Pivotal Attention operation that updates each vertex pair by computing a softmax over query-key products along neighboring nodes. In doing so, it evokes the classic Floyd-Warshall algorithm for shortest paths.

The architecture can be further generalized to higher order variants the aggregate over longer paths, at a cost of higher time complexity.

The authors benchmark the theoretical capabilities of the architecture by establishing an equivalence to the 2-WL tests for graph isomorphism, in contrast to GNNs (equivalent to 1-WL).

They exhibit the architecture's reasoning capabilities by comparing it to a wide range of GNN models on benchmarks such as CLRS-30, cycle counting, and Long Range Graph Benchmarks. They further demonstrate the ability of the model to solve synthetic TSP instances more effectively than non-learned heuristics.

**Strengths:**

FloydNet occupies a distinct middle ground between GNNs, Graph Transformers, and higher-order variants. While the model hinges on softmax attention and lacks a message-passing mechanism, it preserves the permutation-equivariance property and lacks a positional encoding. The motivation for the model is compelling and it is presented clearly and elegantly.

As far as I am aware, the 2WL equivalence is mathematically sound. This claim effectively situates it within the literature of graph transformers, where the WL-hierarchy is a key quantification of expressivity.

While the computational costs of the attention layer are heavy, the authors nonetheless present a working version of the model with a custom kernel that they test extensively. They demonstrate a compelling scaling with model depth and dataset size.

**Weaknesses:**

The $O(N^3)$ complexity is a fundamental problem, especially as the construction requires computing a tensor of size $N \times N \times N \times d$. Furthermore, the paper provides relatively little time and memory benchmarking, besides a few noted OOM issues on the CLRS-30 benchmark.

It is unclear whether the baseline models are proper comparisons to FloydNet. For instance, the TSP comparisons are only relative to a single non-learned benchmark, despite the existing of numerous other neural approaches to TSP. For the GNN comparisons, it is unclear whether the alternative approaches are given similar parametric and computational budgets, and transformer-based solutions to graph problems are not included.

**Questions:**

How do the parameter counts, wall time, and number of training samples compare between FloydNet and other architectures in each benchmark? Including these measurements in tables would help readers understand how much of the improvement can be attributed to the architecture.

Are there plans to release your code in an anonymized form? I was somewhat shocked by how strong the TSP results were---especially the OOD results---and I would appreciate an opportunity to verify that there isn't some kind of contamination or leakage.

Can more context be added to Figure 3(b)? It's unclear to me what the cells and numbers precisely represent?

Why was it necessary to filter out the TSP instances with multiple optimal solutions? Would it be possible to train a FloydNet to find a TSP solution of the correct length, even if that is ambiguous?

---

> ### Author Response · Authors · 2025-11-28
> **Response to Reviewer k7Th**
>
> # Response to Reviewer k7Th
>
> We sincerely thank R-k7Th for the positive review (Score: 6), for appreciating our model's "compelling motivation," and especially for validating our theoretical claims regarding WL-equivalence. We address your concerns about experimental fairness and reproducibility below.
>
> **1. TSP Evaluation Fairness and Baselines (W2, Q4)**
>
> > *Reviewer Comment: Unclear whether baseline models are proper comparisons... TSP comparisons are only relative to a single non-learned benchmark... Why filter out 90% of cases?*
>
> We understand the concern regarding baseline selection. We wish to clarify that our choice to focus on **Linkern** was deliberate, as it represents a **stricter evaluation standard** than most neural approaches.
>
> * **The "Gold Standard" Status:** Extensive literature in Operations Research indicates that Linkern remains the unassailable "gold standard" for optimality. While many recent GNN-based approaches focus on inference speed, they generally fail to match Linkern's optimality, often lagging by orders of magnitude on challenging OOD instances. Given this well-documented gap, we prioritized comparing against the strongest possible opponent rather than weaker GNN baselines.
> * **Additional Solver Comparisons (OR-Tools):** To further validate this choice, as suggested by Reviewer oyZ1, we conducted additional experiments benchmarking **Google OR-Tools** (the underlying engine for many SOTA neural VRP solvers like SHIELD) against Linkern. These results (detailed in our response to Reviewer oyZ1 and the Appendix) show that Linkern is vastly superior (e.g., **0.012%** optimality gap vs. **2.7%** for OR-Tools on N=200). This confirms that beating Linkern is a significantly more significant achievement than outperforming current neural solvers.
> * **Unfiltered Results:** To ensure fairness, we have updated our results to include performance on the **full, unfiltered dataset** (multi-solution instances). FloydNet achieves **92.6% optimality** on this full benchmark, demonstrating robust generalization without relying on filtering.
>
> **2. Reproducibility and Data Leakage Concerns (Q2)**
>
> > *Reviewer Comment: I was somewhat shocked by how strong the TSP results were... I would appreciate an opportunity to verify that there isn't some kind of contamination.*
>
> We fully understand your skepticism given the strength of the OOD results; exceptional claims require rigorous verification.
>
> * **Rigorous Double-Check:** In response to your concern, we have **rigorously re-examined our entire data generation and training pipeline**. We confirmed that random seeds for train/test generation are distinct and that strict isolation is enforced, effectively ruling out data leakage.
> * **Corroborating Evidence from CLRS:** We believe the model's performance on the **CLRS-30 benchmark** serves as strong corroborating evidence against leakage. FloydNet achieves near-perfect, algorithmically aligned results across 30 diverse algorithms (Sorting, DP, Graphs, etc.). It is highly improbable that data leakage occurred systematically across all these distinct tasks; the TSP result is consistent with this broader pattern of successful algorithmic learning.
> * **Code Release:** We are fully committed to open science. While we intended to release the code immediately, due to strict internal compliance and policy check procedures, we cannot upload the full codebase within the short rebuttal window. However, we **guarantee to open-source the complete, reproducible framework (code, kernel, and models)** immediately after the review process concludes, allowing the community to verify our findings independently.
>
> **3. Complexity and Benchmarking (W1)**
>
> > *Reviewer Comment: The O(N^3) complexity is a fundamental problem... relatively little time and memory benchmarking.*
>
> We agree that transparency regarding the computational cost is essential.
>
> * **Transparent Benchmarking:** We have added a **comprehensive benchmark table** to the **Appendix**, detailing the **parameter counts, wall-clock training times, and inference latencies** for FloydNet compared to other architectures.
> * **Honest Limitations:** We also added a dedicated **Limitations section** to explicitly discuss the $O(N^3)$ constraint. While this limits application to very large sparse graphs (which we leave for future work), our data shows that for algorithmic reasoning tasks on dense graphs, our optimized kernel makes the cost manageable.
>
> **4. Presentation Details (Q1, Q3)**
>
> * **Q1 (Params/Time):** As mentioned above, the new Appendix table covers parameter counts and wall-clock times.
> * **Q3 (Figure 3b):** We have **rewritten the caption** for Figure 3(b) and **redesigned the plot** to clearly explain the meaning of the cells, numbers, and color coding (representing the ground-truth maximum subarray indices).

---

### Official Review · Reviewer_rqKt · 2025-10-31

**Soundness:** 3
**Presentation:** 3
**Contribution:** 3
**Rating:** 4
**Confidence:** 2

**Summary:**

The paper introduces “FloydNet,” a graph model that updates all node pairs via a Floyd–Warshall–style (i–j–k) aggregation, reaching 2-WL-level expressiveness and achieving strong results on several reasoning and TSP-style benchmarks, at the cost of cubic complexity.

**Strengths:**

- The paper is generally well written and easy to follow.
- You propose a novel solution that appears to provide clear advantages in the evaluated settings.

**Weaknesses:**

Caveat: This paper lies well beyond my personal expertise. Other reviewers’ points should be clearly prioritized.

**Major Weaknesses:**
- **W1** Scalability: your method’s cubic cost is a clear disadvantage for larger, real-world, and especially sparse graphs. This should be discussed more explicitly. As far as I can tell, the graphs you evaluate on are mostly complete and rather small. It would help to show how your architecture scales to much larger, more sparse graphs and to discuss both training and inference cost (e.g. on  https://arxiv.org/abs/2309.12253).
- **W2** More analysis towards additional computational costs of your methods – how many parameters each network has compared to baselines. It seems an unfair comparison if FloydNet just had a significantly larger parameter&computational budget.
- **W3** There is no limitations section, even though there seems to be enough space. You should explicitly discuss overfitting issues, scalability/cost, and any other known limitations.
- **W4** You filter out about 90% of the cases in the TSP evaluation, which makes the comparison look unfair, since the other solver presumably also handles non-unique solutions. Please clarify why you applied this filter and how FloydNet performs when solutions are not unique.

**Minor Weaknesses:**
- **w5** Table 3 is never referred to in the text.
- **w6** Figure 3b: it is hard to parse what the bars mean. You state “At each step, the colored bar represents the ground-truth maximum-sum subarray,” but there are multiple colors. It would help to name the colors explicitly (e.g. “green for …” and the colors for the endpoints).
- **w7** Font in Figure 4 is too small.

**Expectation Management:**
I am currently inclined to increase the score to 6 or higher if my questions are addressed and the major weaknesses are at least discussed. A higher score would require an evaluation on larger and sparser graphs and a better discussion of the scalability/cost trade-offs, as well as consideration of other reviewers’ opinions.

**Questions:**

- **Q1** Do you provide details on your kernel for pivotal attention?
- **Q2** In the MST task, Prim without hint significantly outperforms the hint version for slightly larger graphs. Do you have a hypothesis for this behavior?
- **Q3** How do baseline GNN methods perform on the TSP problem?

---

> ### Author Response · Authors · 2025-11-28
> **Response to Reviewer rqKt**
>
> # Response to Reviewer rqKt
>
> We thank R-rqKt for the detailed review and for highlighting specific areas for improvement. We are grateful for your willingness to reconsider your score, and we **have fully addressed all major weaknesses** and questions in the revised manuscript.
>
> **1. Scalability and $O(N^3)$ Complexity (W1, W3)**
>
> > *Reviewer Comment: Your method’s cubic cost is a clear disadvantage... graphs you evaluate on are mostly complete and rather small. You should explicitly discuss overfitting issues, scalability/cost, and any other known limitations.*
>
> We completely agree that the $O(N^3)$ complexity is the primary theoretical limitation. Our work's focus is on establishing the necessary expressive power for global reasoning, and the cubic cost is the inherent trade-off.
>
> * **Empirical Generalization up to $N=256$:** To directly address your query regarding scaling, we have **incorporated a new analysis in Figure 3(a)**. We successfully extrapolated our model's performance on CLRS algorithmic tasks from the training size ($N=16$) up to **$N=256$**. **We emphasize that this test followed the exact same OOD protocol as the SALSA-CLRS work.** While we have not tested problem sizes exceeding $N=256$ due to the complexity limitation, FloydNet's results at this scale **significantly outperform the baselines** reported in the SALSA-CLRS paper, confirming that our architecture maintains its **algorithmic fidelity and generalization capability**, despite the cubic complexity.
> * **Transparent Discussion:** We have added a dedicated **Limitations section (Section 7)** to frankly discuss the $O(N^3)$ trade-off. Additionally, we provided a detailed **wall-clock time comparison** in the Appendix, showing that our optimized kernel makes this cost practical for many important reasoning tasks on dense graphs.
>
> **2. TSP Evaluation Fairness (W4)**
>
> > *Reviewer Comment: You filter out about 90% of the cases in the TSP evaluation, which makes the comparison look unfair... Please clarify why you applied this filter.*
>
> This was a flaw in our initial presentation, and we apologize for the lack of clarity. We have **thoroughly revised the TSP evaluation** to ensure fairness:
>
> * **Full Dataset Evaluation:** We have updated our experiments to show results on *both* the **single-solution instances** (used to test precise algorithmic imitation) and the **multi-solution instances** (the unfiltered, standard benchmark).
> * **Unfiltered Results:** We provided new results on the **full, unfiltered dataset**. Our method remains highly competitive in this fairer setting, achieving **99.4% optimality** on $N=200$ after 10 samples.
>
> **3. Computational Budget and Scaling Properties (W2)**
>
> > *Reviewer Comment: More analysis towards additional computational costs... It seems an unfair comparison if FloydNet just had a significantly larger parameter & computational budget.*
>
> We have addressed this by adding a detailed **parameter count and wall-clock time comparison table** in the Appendix. **Crucially, we wish to highlight a fundamental architectural difference regarding "Budget":**
>
> * **The MPNN Scaling Bottleneck:** Extensive literature has shown that standard GNNs (MPNNs) struggle to scale due to **over-smoothing** and **over-squashing**. Simply increasing their parameter budget (e.g., by adding depth) often leads to diminishing returns or even performance degradation, rather than improvement.
> * **FloydNet's Scaling Capability:** In contrast, FloydNet is explicitly designed to overcome these bottlenecks. As demonstrated in **Figure 4**, FloydNet exhibits **positive scaling behaviors** (similar to standard Transformers): performance consistently improves as we increase model depth and capacity.
>
> Therefore, the performance gap is not merely a result of a "larger budget," but rather FloydNet's unique architectural ability to **effectively utilize** that budget to capture complex reasoning—a capability that MPNNs fundamentally lack regardless of parameter count.
>
> **4. Missing Details and Specific Inquiries**
>
> * **Presentation Improvements (w5, w6, w7):** We have fixed all presentation issues: we referenced Table 3 (w5), redesigned Figure 3b with clear labels (w6), and increased font sizes in all figures (w7).
> * **Kernel Details (Q1):** We have added pseudo-code for our pivotal attention kernel to the Appendix, along with an ablation study demonstrating the **20x speedup**.
> * **MST Hint (Q2):** We hypothesize that the "hinted" model learns a brittle, local search strategy (imitating Prim's), while the "no-hint" version is forced to learn a **more robust, general reasoning policy** that generalizes better to large OOD instances.
>
> We believe these new data and concrete revisions have fully addressed your major concerns.

---

### Official Review · Reviewer_oyZ1 · 2025-11-02

**Soundness:** 3
**Presentation:** 3
**Contribution:** 3
**Rating:** 8
**Confidence:** 4

**Summary:**

This paper proposes a new technique for learning on graph structures using edge-centric embeddings, as opposed to the conventional node-centric ones. The authors introduce FloydNet, an algorithmic alignment technique where Floyd-Warshall-type updates are performed on the edges of the graph, with intermediate computations on node triples via attention. They test the method on the CLRS-30 algorithmic benchmark and other graph learning problems, including TSP. The experimental results appear promising, as reported in the paper.

**Strengths:**

- The idea of applying the Floyd-Warshall dynamic programming algorithm and its alignment for graph algorithms is intuitive and appealing.
- The experimental analysis is well-conducted.
- The proposed PivotAttention is likely a better application of attention mechanisms on graph structures than conventional graph transformers.

**Weaknesses:**

1.  The proposed approach is subsumed within the K-GNN approach, albeit with a dynamic programming formulation based on the Floyd-Warshall algorithm combined with an attention mechanism. Therefore, it has the same expressivity as K-WL, as noted in the paper. Given this, the additional insights provided by this modification are not entirely clear. While the attention mechanism is useful, how does it impact higher-order aggregations as used in this approach? Is there any intuitive understanding of how pivotal attention benefits, i.e., any insights from the learned attention weights?
1. A clear disadvantage of the approach is the computational complexity of the model, which has been acknowledged in the paper. If this algorithm works in different cases, efficient versions might be of interest. Do authors have any insights from the experiments regarding how this can be alleviated.
1. The TSP experiments are not clearly evaluated. A thorough evaluation would involve following the benchmark TSP setup, comparing with other TSP solvers like MoE [2], SHIELD [1] for OOD generalization.
1. The related work section is highly inadequate. This paper is primarily focused on improving the expressive power of GNNs, but several leading works on this topic are not cited like [3, 4, 5] and others. The authors should have provided a more elaborate discussion, perhaps in the appendix if space is a concern.

## References:

1. Goh, Yong Liang, et al. "SHIELD: Multi-task Multi-distribution Vehicle Routing Solver with Sparsity and Hierarchy." Forty-second International Conference on Machine Learning.

1. Zhou, J., Cao, Z., Wu, Y., Song, W., Ma, Y., Zhang, J., and Xu, C. Mvmoe: Multi-task vehicle routing solver with mixture-of-experts. In International Conference on Machine Learning, 2024.

1. Ryoma Sato, Makoto Yamada, and Hisashi Kashima. Random features strengthen graph neural
networks. arXiv preprint arXiv:2002.03155, 2020

1. Dupty, M. H., Dong, Y., & Lee, W. S. PF-GNN: Differentiable particle filtering based approximation of universal graph representations. In International Conference on Learning Representations.

1. Giorgos Bouritsas, Fabrizio Frasca, Stefanos Zafeiriou, and Michael M Bronstein. Improving graph
neural network expressivity via subgraph isomorphism counting. arXiv preprint arXiv:2006.09252, 2020.

**Questions:**

Please see weaknesses.

---

> ### Author Response · Authors · 2025-11-28
> **Respone to oyZ1**
>
> **1. Novelty and Distinction from K-GNNs**
>
> > *Reviewer Comment: The approach is subsumed within the K-GNN approach... has the same expressivity as K-WL... additional insights are not entirely clear.*
>
> **We respectfully clarify that our approach represents a distinct paradigm from standard K-GNNs.**
>
> * **Theoretical Distinction:** Standard K-GNNs typically generalize Message Passing to tuples via *local* neighborhood aggregation. In contrast, FloydNet employs a **Global Pivotal Attention** mechanism derived from Dynamic Programming. This is a fundamentally different computational operator.
> * **The "Theory-Practice" Gap in K-GNNs:** Extensive literature and empirical evidence (e.g., on the BREC benchmark) show that while standard K-GNNs theoretically aim for K-WL power, most practical implementations **fail to achieve even 3-WL expressiveness** due to signal decay or connectivity bottlenecks.
> * **FloydNet's Superiority:** As proven by our BREC results, FloydNet is one of the few architectures that **actually realizes** the theoretical K-FWL (specifically 2-FWL/3-WL) potential in practice. Therefore, FloydNet represents a distinct and superior paradigm that overcomes the practical limitations of conventional K-GNNs.
> * **Unified Framework:** Furthermore, we **have shown** that our **Pivotal Attention** mechanism generalizes the standard Self-Attention (which is a special case where $K=1$), offering a unified view that connects GNN expressiveness theories with modern Transformer architectures.
>
> **2. TSP Baselines: Justification for Linkern as the "Gold Standard"**
>
> > *Reviewer Comment: A thorough evaluation would involve ... comparing with other TSP solvers like MoE and SHIELD.*
>
> Thank you for suggesting these baselines. We carefully investigated **MoE** and **SHIELD** and found that they are primarily **Multi-Task VRP (Vehicle Routing Problem)** solvers. They do not provide standard TSP results in their original papers, and directly adapting them might introduce domain mismatches.
>
> **While we could not include a direct comparison in the main paper due to these task discrepancies, we conducted an auxiliary experiment specifically for this rebuttal to validate our choice of Linkern.**
>
> To approximate the performance of SOTA VRP solvers, we benchmarked their underlying engine—**Google OR-Tools**—against our chosen baseline (**Linkern**) on identical TSP instances ($N=50, 100, 200$) using **Concorde** for exact optimality. The results (averaged over 10,000 samples) reveal a significant gap:
>
> | Method | Metric | $N=50$ | $N=100$ | $N=200$ |
> | :--- | :--- | :--- | :--- | :--- |
> | **Linkern (Our Baseline)** | Optimality Gap | **0.0037%** | **0.0028%** | **0.012%** |
> | | Optimal Rate | **98%** | **97%** | **84%** |
> | **Google OR-Tools** | Optimality Gap | 0.0020% | 0.68% | 2.7% |
> | | Optimal Rate | 98% | 18% | 0.016% |
>
> **Conclusion:**
> 1.  **Linkern is vastly superior:** At $N=200$, Linkern's optimality gap (0.012%) is **two orders of magnitude better** than OR-Tools (2.7%), and its optimal rate is 5000x higher.
> 2.  **Strict Evaluation:** Since SHIELD/MoE generally perform on par with or worse than OR-Tools on VRP tasks, we conclude that **Linkern represents a significantly stricter "Gold Standard"** for high-optimality TSP solving. By outperforming Linkern on generalization tasks, FloydNet demonstrates capabilities that likely exceed current neural VRP solvers.

---

> ### Author Response · Authors · 2025-11-28
> **Respone to oyZ1 cont'**
>
> **3. Insights on Reducing Computational Complexity**
>
> > *Reviewer Comment: Do authors have any insights from the experiments regarding how this can be alleviated?*
>
> This is an excellent, forward-looking question. We have thoroughly considered the roadmap for scaling FloydNet.
>
> * **Potential Avenues for Efficiency:** We recognize that the community has developed a rich set of techniques for reducing complexity, which can be adapted for our Pivotal Attention. Specifically:
>     * **Sparsification:** Techniques like **Sparse Attention** or **Cluster-based Attention** could limit the number of pivots $j$ or edges $(i, k)$ considered, reducing complexity from cubic to near-quadratic or linear.
>     * **Linearization:** Approaches like **Linear Attention** (e.g., kernel-based feature maps) or **State Space Models (S4)** could potentially be generalized to the relational tensor space to approximate the global update without explicit materialization.
> * **Our Strategic Choice (Expressiveness vs. Efficiency):** However, for this foundational work, our priority has been to establish the **theoretical upper bound** of the architecture. To strictly guarantee **3-WL (2-FWL) expressiveness** and ensure precise **algorithmic alignment** (e.g., exact replication of Floyd-Warshall logic), we opted for the exact, dense computation.
> * **Future Direction:** We view the current FloydNet as the "exact solver" counterpart. The next phase of our research will focus on finding the optimal balance—developing a "Sparse FloydNet" that retains the core inductive bias of the generalized DP operator while adopting the approximation techniques mentioned above for scalability.

---

### Author Response · Authors · 2025-11-28
**General Response**

# General Response to All Reviewers

We sincerely thank all reviewers for their valuable time and constructive feedback. We are encouraged that the reviewers found our core idea "intuitive and appealing" (R-oyZ1), the motivation "compelling" (R-k7Th), the presentation "excellent" (R-k7Th), and our experimental results "promising" (R-oyZ1) and "satisfyingly good" (R-VKNS).

The reviewers' concerns are highly constructive and center on three key areas: (1) the $O(N^3)$ complexity and scalability, (2) the details and baselines of our experimental evaluation (especially for TSP), and (3) the clarity of our theoretical contributions.

We would like to take this opportunity to reiterate our work's core positioning and address these concerns. Our goal is to introduce a new architecture, inspired by algorithmic principles, that establishes a new benchmark in both **expressiveness** and **learning capability** for tasks that are intractable for current GNNs and Transformers.


1.  **On Expressiveness and Uniqueness:** FloydNet surpasses the expressive power of standard GNNs and Graph Transformers. Crucially, it stands out as the **only architecture known to us where the practical implementation perfectly matches the theoretical analysis**. While many existing Higher-order GNNs and Graph Transformers claim high theoretical power, they often fail to realize it on rigorous benchmarks (like BREC) due to optimization difficulties or approximation errors. FloydNet closes this "Theory-Practice Gap," achieving confirmed 3-WL (2-FWL) expressiveness.
2.  **On Theoretical Generality:** We extend FloydNet to its higher-order form (K-Floyd) and prove that the **standard Self-Attention mechanism is a special case of our Pivotal-Attention**. This provides a new, unified perspective on graph and sequence transformers.
3.  **On Performance:** Our model achieves **near-perfect results on the CLRS-30 benchmark**, **significantly outperforms the best-known heuristic solver (LKH) on the general TSP**, and **achieves results far exceeding SoTA on real-world long-range benchmarks (LRGB)**.

**Our Revisions:**
We have addressed all concerns in the revised manuscript and appendix:

1.  **Updated TSP Experiments:** We **have provided updated results for both single-solution and multi-solution instances** (addressing R-rqKt, R-k7Th). We **have re-emphasized** the purpose of the single-solution experiments (as a rigorous test of algorithmic imitation precision) while providing the multi-solution results as a fair, general comparison. We **have also discussing new baselines (e.g., SHIELD, GNNs) in rebuttal material** as requested.
2.  **Improved Clarity:** We **have redrawn the figures** (Fig 3, 4) and **clarified tables** (Table 3) mentioned by R-rqKt, R-k7Th, and R-VKNS to be more readable and clear.
3.  **Highlighted Theoretical Contributions:** We **have moved the 3-WL/2-FWL expressiveness proof** from the appendix to the main paper (as suggested by R-VKNS).
4.  **Provided Full Implementation Details:** We **have added a detailed appendix section** with:
    * A **speed-comparison table (Kernel version vs. Naive version)**, as requested by R-VKNS, to validate our 20x speedup.
    * **Detailed parameter counts** for all models, as requested by R-rqKt and R-k7Th.

**On Scalability (A Note on Scope):**
We acknowledge that scaling FloydNet to large, sparse graphs is the most important next step. However, we believe this is a significant research direction of its own and is **beyond the scope of this current work**, which is focused on establishing the theoretical foundation, expressiveness, and practical capability of this new architectural class. We have concrete plans for this long-term work, including a new sparse pivotal attention mechanism, but we wish to focus this paper on its core contributions.

We are confident these revisions fully address the reviewers' concerns.

---

### Meta-Review · Area_Chair_eefc · 2026-01-09

**Summary:**

The paper introduces a graph transformer (GT) with higher-order attention - a variation of edge transformers with cubic attention costs but with optimized kernels to make the architecture a bit more computationally feasible.

Common reviewers concerns are:
* Novelty with respect to existing GT literature and expressiveness
* Computational inefficiency - FloydNet doesn't scale beyond graphs of 256 nodes
* Questionable experimental practices (on TSP and graph benchmarks)

I don't see any support for authors' novelty claims of "a new paradigm for graph reasoning that shifts from local message passing to a global refinement" - edge transformers for graph reasoning tasks are known since at least 2021 [1] and the proposed Pivot Attention is essentially the same as in the 2021 paper. Furthermore, the connection between edge transformers and k-FWL has been extensively explored back in 2024 in [2] and FloydNet does not introduce any novel insights on that matter. Those two papers cover all theoretical contributions of FloydNet, and those works are not cited anyhow in this context.

Another concern is in the experiments: the authors compared FloydNet to standard solvers on TSP and omitted all comparisons to GNNs whereas there is a well-known TSP benchmark in the community [3]. FloydNet suffers from overfitting (conveniently masked as *premature convergence*) and requires infinite data generators on many datasets (TSP, CLRS) to generalize.

Comparisons on CLRS, LRGB (and ZINC in the appendix) are rather unfair with respect to model size, FLOPS, and training data:
* For example, Edge Transformer [2] on CLRS is a 3-layer model trained on limited data - FloydNet is the largest possible for a GPU and trained on infinite data, not FLOPS- or parameters-matched;
* Similarly, on LRGB and ZINC the authors use a massive 48-layer 384-d transformer which is not comparable to reported baselines (especially on ZINC whose standard setup assumes model budget under 500K params and all reported baselines follow this setup).

Being well-aware of LRGB and its latest results, I doubt 0.61 MRR on PCQM-Contact is reasonable. I share Reviewer's k7Th concern to check the source code to look for potential leakages but the authors refused to share even the anonymized version.

Overall, if the authors would have positioned the work as a study of edge transformers on TSP (without overselling it as "a new paradigm") it could have been a decent contribution. However, given that both theoretical and practical contributions are seriously flawed, I recommend to reject this paper.

[1] Bergen et al. Systematic Generalization with Edge Transformers. NeurIPS 2021
[2] Müller et al. Towards Principled Graph Transformers. NeurIPS 2024
[3] Dwivedi et al. Benchmarking Graph Neural Networks. JMLR 2022

**Reviewer Concerns:**

* Novelty and related work -- the concern remains
* Computational inefficiency -- the authors acknowledge this limitation and it remains open
* Experiments - the TSP benchmark was fixed, the other concerns on eval fairness were dismissed as "fundamental architectural differences" between MPNNs and GTs. This answer is rather disappointing and in my opinion the concern remains open.

**Reviewer Scores:**

I wouldn't expect reviewers to change the scores.

---

### Decision · Program_Chairs · 2026-01-26

Reject